# DATA DRIFT CORRECTION VIA TIME-VARYING IMPORTANCE WEIGHT ESTIMATOR

## ABSTRACT

Real-world deployment of machine learning models is challenging when data evolves over time. And data does evolve over time. While no model can work when data evolves in an arbitrary fashion, if there is some pattern to these changes, we might be able to design methods to address it. This paper addresses situations when data evolves gradually. We introduce a novel time-varying importance weight estimator that can detect gradual shifts in the distribution of data. Such an importance weight estimator allows the training method to selectively sample past data—not just similar data from the past like a standard importance weight estimator would but also data that evolved in a similar fashion in the past. Our time-varying importance weight is quite general. We demonstrate different ways of implementing it that exploit some known structure in the evolution of data. We demonstrate and evaluate this approach on a variety of problems ranging from supervised learning tasks (multiple image classification datasets) where the data undergoes a sequence of gradual shifts of our design to reinforcement learning tasks (robotic manipulation and continuous control) where data undergoes a shift organically as the policy or the task changes.

## 1 INTRODUCTION

Real-world machine learning performance often drops during deployment when test data no longer stem from the same distribution from which previous training data were sampled. Thus many tools have been developed to address data distribution shift[1] (Heckman, 1979; Shimodaira, 2000; Huang et al., 2006; Bickel et al., 2007; Sugiyama et al., 2007b; 2008; Gretton et al., 2008), often using an estimate of the Radon-Nikodym (Rosenbaum & Rubin, 1983) derivative between the two distributions (also known as a *propensity score* or *importance weight*) to re-weight the training data so that its weighted distribution better matches the test data (Agarwal et al., 2011; Wen et al., 2014; Reddi et al., 2015b; Chen et al., 2016; Fakoor et al., 2020b;a; Tibshirani et al., 2019). These methods mostly consider offline settings with one training and one test dataset.

In many applications, data for supervised learning is continuously collected from a constantly evolving distribution (e.g. due to a pandemic (Callaway, 2020)) such that the single train/test paradigm no longer applies. Settings in which the data distribution drifts gradually over time (rather than experiencing abrupt changes) are particularly ubiquitous (Shalev-Shwartz, 2012). Assuming observations are not statistically dependent over time (as in time-series), how to best train models in such streaming/online learning settings remains a key question (Lu et al., 2018). Some basic options include: fitting/updating the model to only the recent data (which is statistically suboptimal if the distribution has not significantly shifted) or fitting the model to all previous observations (which leads to biased estimates if shift has occurred). Here we consider a less crude approach in which past data are weighted during training with continuous-valued weights that vary over time. Our proposed estimator of these weights generalizes standard two-sample propensity scores, allowing the training process to selectively emphasize past data collected at time $t$ based on the distributional similarity between the present and time $t$.

---

[1]Covariate, label, and concept shifts are all referred to as distribution shift in the paper to simplify things wherever it is clear from the context.

We evaluate our proposed method in supervised and reinforcement learning settings involving a sequence of gradually changing tasks with slow repeating patterns. In such settings, the model not only must continuously adapt to changes in the environment/task but also learn how to select past data which may have become more relevant for the current task. Comprehensive experiments demonstrate that our method can effectively detect shift in data and statistically account for such changes during learning and inference time.

## 2 RELATED WORK

Given its importance in real-world applications, the problem of how to learn from shifting distributions has been widely studied. Much past work has focused on a single shift between training/test data (Lu et al., 2021; Wang & Deng, 2018; Fakoor et al., 2020b) as well as restricted forms of shift involving changes in only the features (Sugiyama et al., 2007a; Reddi et al., 2015a), labels (Lipton et al., 2018; Garg et al., 2020; Alexandari et al., 2020), or in the underlying relationship between the two (Zhang et al., 2013; Lu et al., 2018). Past approaches to handle distributions evolving over time have been considered in the literature on: concept drift Gomes et al. (2019); Souza et al. (2020), reinforcement learning (shift between the target policy and behavior policy) Schulman et al. (2015); Wang et al. (2016); Fakoor et al. (2020a), (meta) online learning Shalev-Shwartz (2012); Finn et al. (2019); Harrison et al. (2020); Wu et al. (2021), and task-free continual/incremental learning Aljundi et al. (2019); He et al. (2019), but to our knowledge, existing methods for these settings do not employ time-varying data weights like we propose here. Time-varying data weights have been considered in survival analysis which models longitudinal observations subject to censoring Lu (2005); Cox (1972), but our weights here are implemented differently (via deep learned estimates) and utilized for general supervised learning and reinforcement learning with drift.

## 3 APPROACH

Consider a standard learning problem where training data are drawn from a probability distribution $p(x)$ and test data have a different probability distribution $q(x)$. Our goal is to build a model that can predict equally well on the training and test data distributions. We can do so by observing that

$$\mathbb{E}_{x \sim q(x)}[\ell(x)] = \int \frac{\mathrm{d}p(x)}{\mathrm{d}p(x)} \, \mathrm{d}q\,(x)\ell(x) = \mathbb{E}_{x \sim p(x)}\Big[\frac{\mathrm{d}q(x)}{\mathrm{d}p(x)}\ell(x)\Big] = \mathbb{E}_{x \sim p(x)}\Big[\beta(x)\,\ell(x)\Big] \quad (1)$$

where $\ell(x)$ is any function, say the loss of our model. The propensity score $\beta(x) = \frac{\mathrm{d}q(x)}{\mathrm{d}p(x)}$ is the importance ratio; this can also be seen as the Radon-Nikodym derivative (Resnick, 2013) of the two distributions. The propensity score measures the likelihood that a sample $x$ came from distribution $p$ against it coming from a distribution $q$. We will call $\beta(x)$ the "standard propensity score".

Since densities $q(x)$ and $p(x)$ are often unknown, a binary classifier (Agarwal et al., 2011) needs to be used to estimate $\beta(x)$ by utilizing samples drawn from $p$ and $q$. In particular, we want to create a data set, $D = \{(x_i, z_i)\}_{i=1}^N$ for the binary classifier such that $z_i = 1$ if $x_i \sim p$ and $z_i = -1$ if $x_i \sim q$. We want our training data to have $z = 1$ in half of the training examples (with $x \sim p$) and $z = -1$ (with $x \sim q$) in the other half. We fit the binary classifier by solving the following:

$$\underset{\theta}{\text{minimize}} \ \frac{1}{N} \sum_{(x_i, z_i)}^{N} \log\Big(1 + \exp(-z_i g_\theta(x_i))\Big) \quad (2)$$

where $g_\theta$ can be either a linear or non-linear model parameterized by $\theta$.

### 3.1 TIME-VARYING IMPORTANCE WEIGHT ESTIMATOR

Let us now consider time-varying probability distribution $p_t(x)$ with $t = 1, \ldots, T$. We will assume that we have recorded tuples $D = \{(x_i, t_i)\}_{i=1}^m$ with each $x_i \sim p_{t_i}(x)$. Our goal is to build a model that can predict well on data from $p_T(x)$ using the historical data $D$, say we seek to minimize

$$\mathbb{E}_{x \sim p_{T+\mathrm{d}t}(x)}[\ell(x)]. \quad (3)$$

Observe that we seek to build a model not for data from $p_T$ but instead from $p_{T+\mathrm{d}t}$ where we will think of $\mathrm{d}t$ as small step in the future. **This problem is challenging because we do not have data from $p_{T+\mathrm{d}t}$.**

Let us define $p(t) = N^{-1} \sum_{i=1}^{N} \delta_{\{t=t_i\}}$ which is simply the empirical distribution of the time instants at which we received our samples; here $\delta$ is the Dirac-delta distribution. We can now define the marginal on the data as

$$p(x) = \int_0^T \mathrm{d}t\, p(t)\, p_t(x).$$

Inspired by the above expression, we will liken $p_t(x) \equiv p(x \mid t)$.

**Assumption 1 (Data evolves gradually).** If data evolves arbitrarily, then it is difficult to expect a model learned from past data to be able to predict accurately on future data. But observe that

$$\mathbb{E}_{x \sim p_{T+\mathrm{d}t}(x)} [\ell(x)] = \mathbb{E}_{x \sim p_T(x)} [\ell(x)] + \int \mathrm{d}x\, (p_{T+\mathrm{d}t}(x) - p_T(x))\, \ell(x) \tag{4}$$
$$\leq \mathbb{E}_{x \sim p_T(x)} [\ell(x)] + 2\mathrm{TV}(p_{T+\mathrm{d}t}(x), p_T(x)),$$

where $\mathrm{TV}(p_{T+\mathrm{d}t}(x), p_T(x)) = \frac{1}{2} \int \mathrm{d}x\, |p_{T+\mathrm{d}t}(x) - p_T(x)|$ is the total variational divergence between the probability distributions at time $T$ and $T + \mathrm{d}t$, and the integrand $\ell(x)$ is upper bounded by $1$ in magnitude (without loss of generality, say after normalizing it). We therefore assume that the changes in our time-varying probability distribution $p_T(x)$ are upper bounded by a constant (uniformly over time $t$); this can allow us to build a model for $p_{T+\mathrm{d}t}$ using data from $\{p_t\}_{t \leq T}$.

**Validating and formalizing Assumption 1** This assumption is a natural one in many settings and we can check it in practice using a two-sample test to estimate the total variation $\mathrm{TV}(p_{T+\mathrm{d}t}, p_T)$ (Gretton et al., 2012). We do so for some of our tasks in §4 (see Fig. 2).We can also formalize this assumption mathematically as follows. The transition function $P_t$ of a Markov process $X_t$ satisfies

$$\mathbb{E}\left[\varphi(X_t)\right] = \int_0^t \varphi(y) P_t(X_s, \mathrm{d}y).$$

for any measurable function $\varphi$. The distribution of the random variable $X_t$ corresponding to this Markov process is exactly our $p_t$. We can define the semi-group of this transition function as the conditional expectation $K_t \varphi = \int \mathrm{d}y\, \varphi(y) P_t(x, \mathrm{d}y)$ for any test function $\varphi$. Such a semi-group satisfies properties that are familiar from Markov chains in finite state-spaces, e.g., $K_{t+s} = K_t K_s$. The infinitesimal generator of the semi-group $K_t$ is defined as $A\varphi = \lim_{t \to 0}(\varphi - K_t \varphi)/t$. This generator completely defines the local evolution of a Markov process (Feller, 2008)

$$p_t = e^{-tA}\, p_0 \tag{5}$$

where $e^{tA}$ is the matrix exponential of the generator $A$. If all eigenvalues of $A$ are positive, then the Markov process has a steady state distribution, i.e., the distribution of our data $p_t$ stops evolving (this could happen at very large times and the time-scale depends upon the smallest non-zero eigenvalue of $A$). On the other hand, if the magnitude of the eigenvalues of $A$ is upper bounded by a small constant, then the stochastic process evolves gradually. Note it is not necessary for the stochastic process to have a steady state distribution for our time-varying importance weight to be meaningful. Our method is expected to be effective so long as the process evolves gradually, which most processes do.

**Remark 2 (Evaluating the model learned from data $\{p_t(x)\}_{t \leq T}$ upon test data from $p_{T+\mathrm{d}t}(x)$).** We are interested in making predictions on future data, i.e., data from $p_{T+\mathrm{d}t}$. For all our experiments, we will therefore evaluate on test data from "one time-step in the future". The learner does not have access to any training data from $p_{T+\mathrm{d}t}$ in our experiments. This is reasonable if the data evolves gradually. Note our setting is different from typical implementations of online meta-learning (Finn et al., 2019; Harrison et al., 2020) and continual learning (Hadsell et al., 2020; Caccia et al., 2020).

In the sequel, motivated by Assumption 1, we will build a time-dependent propensity score for $p_T$ instead of $p_{T+\mathrm{d}t}$. Using a similar calculation as that of (1) (see Appendix A for a derivation), we can

write our objective as

$$\mathbb{E}_{x \sim p_T(x)} \left[ \ell(x) \right] = \mathbb{E}_{t \sim p(t)} \mathbb{E}_{x \sim p_t(x)} \left[ \omega(x, T, t) \, \ell(x) \right], \tag{6}$$

where

$$\omega(x, T, t) = \frac{\mathrm{d}p_T}{\mathrm{d}p_t}(x) \tag{7}$$

is a time-varying importance weight estimator.

**Remark 3 (Is the time-varying importance weight equivalent to standard propensity score on $(x, t)$?).** One may be inclined to consider whether the development above remains meaningful if we define a new random variable $z \equiv (x, t)$ with a corresponding distribution $p(z) \equiv p_t(x)p(t)$. However, doing so is not useful for estimating $\mathbb{E}_{x \sim p_T(x)} \left[ \ell(x) \right]$ because our objective (6) involves conditioning on a particular time $T$.

We again need to estimate this score using samples from our dataset $D$. To do so, we will consider two models for $p_t(x)$.

**Method 1: Modeling $p_t(x)$ using an exponential family.**  We model the distribution of $x$ and $t$ as

$$p_t(x) = p_0(x) \exp\left( g_\theta(x, t) \right). \tag{8}$$

for some continuous function $g_\theta(x, t)$ parameterized by weights $\theta$. This gives

$$\omega(x, T, t) = \exp\left( g_\theta(x, T) - g_\theta(x, t) \right) \tag{9}$$

Now observe that the left-hand side can be calculated using the same objective that we had in (2) to estimate $g_\theta(x, t)$. To train the binary classifier, we need to create a triplet $(x_i, t', t)$ for each of samples in the dataset with label 1 if $t'$ equals the time corresponding to $x_i$, i.e., if $t_i = t'$ and label $-1$ if $t$ equals the time corresponding to $x_i$, i.e. $t_i = t$. These steps are detailed in Algorithm 1.

**Method 2: Modeling deviations from the marginal**  Observe that in the above method $p_0(x)$ is essentially undefined. We can exploit this "gauge" to choose $p_0(x)$ differently. We can model $p_t(x)$ as deviations from the marginal on $x$:

$$p_t(x) = p(x) \exp\left( g_\theta(x, t) \right). \tag{10}$$

This again gives $\omega(x, T, t) = \exp\left( g_\theta(x, T) - g_\theta(x, t) \right)$. For each of the $N$ samples in the dataset, we create a datum $(x_i, t_i)$ with label 1 and another datum $(x_i, t_j)$ for $j \neq i$ with label -1. The logistic regression in (2) is now fitted for the classifier $g_\theta(x, t)$ on this dataset.

The difference between the two methods is subtle. In Method 1, the classifier $g_\theta(x, t)$ is fitted on tuples $(x, t', t)$ while in Method 2, the dataset only consists of tuples $(x, t)$. In the former case, we think of the fitted model $g_\theta(x, t)$ as an estimate of $\log(p(x, t)/p(x, 0))$ while in the latter case we think of the fitted model as the estimate of $\log(p(x, t)/p(x))$. In both cases, we can use this function $g_\theta(x, t)$ to calculate our time-varying importance weight. Because of this, we expect the two methods to perform similarly, as our experiments will also show.

## 3.2 Learning models using data that evolves across time

We next discuss a learning theoretic perspective on how we should learn models when the underlying data evolves with time. For the purposes of this section, suppose we have $m$ input-output pairs $D_t = \{(x_i, y_i)\}_{i=1}^m$ that are independent and identically distributed from $p_t(x, y)$ for $t = 1, \ldots, T$ (note that these are data from each time instant, there can be correlations across time). We seek to learn hypotheses $\overline{h} = (h_t)_{t=1,\ldots,T} \in H^T$ using these samples that can generalize on new samples from distributions $(p_t)_{t=1,\ldots,T}$ respectively. We have denoted the joint hypothesis space as $H^T$. A typical uniform convergence bound on the population risk of one $h_t$ suggests that if

$$m = \mathcal{O}\left( \epsilon^{-2}(\mathrm{VC}_H - \log \delta) \right),$$

then $e_t(h_t) \leq \hat{e}_t(h_t) + \epsilon$ where $e_t(h_t)$ the population risk of $h_t$ on $p_t$ and $\hat{e}_t(h_t)$ is its average empirical risk on the $m$ samples from $D_t$ (Vapnik, 1998). Here VC is the VC-dimension of $H \ni h_t$.

Consider the setting when we want hypotheses that have a small population risk $e^t(\overline{h}) = T^{-1} \sum_{t=1}^T e_t(h_t)$. To achieve this, we may minimize the empirical risk $\hat{e}^t(\overline{h}) = T^{-1} \sum_{t=1}^T \hat{e}_t(h_t)$. Baxter (2000) shows that if

$$m = \mathcal{O}\left(\epsilon^{-2}\left(\text{VC}_H(T) - T^{-1}\log\delta\right)\right) \tag{11}$$

then we have $e^t(\overline{h}) \leq \hat{e}^t(\overline{h}) + \epsilon$ for any $\overline{h}$. The quantity $\text{VC}_H(T)$ is a generalized VC-dimension that characterizes the number of distinct predictions made by the $T$ different hypotheses; it depends upon the stochastic process underlying our individual distributions $p_t$. Larger the time $T$, smaller the $\text{VC}_H(T)$. Whether we should use the $m$ samples from $p_T$ to train one hypothesis, or use all the $mT$ samples to achieve a small *average* population risk across all the tasks, depends upon how related the tasks are. If our goal is only the latter, then we can train on data from all the tasks because

$$\text{VC}_H(T) \leq \text{VC}_H, \text{ for all } T \geq 1.$$

This is fundamentally because all the tasks are together used by the learner to learn an inductive bias, and effectively a smaller hypothesis space to fit the hypotheses $\overline{h}$ from. Such a procedure is effective if tasks are related to each other and there are a number of theoretical and empirical procedures to estimate such relatedness; see Baxter (2000); Gao & Chaudhari (2021); Achille et al. (2019) among others. This result is often used as the motivation for training on multiple tasks; it motivates one of our baselines (**Everything**). The procedure that only builds the hypothesis using data from $p_T$ is our second baseline (**Recent**).

Our goal is however not to obtain a small population risk on all tasks, it is instead to obtain a small risk on $p_{T+dt}$—our proxy for it being the latest task $p_T$. Even if the average risk on all tasks of the hypotheses trained above is small, the risk on a particular task $p_T$ can be large. Ben-David & Schuller (2003) studies this phenomena. They show that if we can find a hypothesis $h \circ f_t$ such that $h \circ f_t$ can predict accurately on $p_t$ for all $t$, then this effectively reduces the size of the hypothesis space $\overline{h} \in H^T$ and thereby entails a smaller sample complexity in (11) at the added cost of searching for the best hypothesis $f_t \in F$ for each task $p_t$ (which requires additional samples that scale linearly with $\text{VC}_F$). The sample complexity of such a two-stage search can be better than training on all tasks, or training on $p_T$ in isolation, under certain cases, e.g., if $\text{VC}_H \geq \log|F|$ for a finite hypothesis space $F$ (Ben-David & Schuller, 2003).

While the procedure that combines such hypotheses to get $h \circ f_t$ in the above theory is difficult to implement in practice, this argument motivates our third baseline (**Finetune**) which first fits a hypothesis on all past tasks $(p_t)_{t \leq T}$ and then adapts it further using data from $p_T$.

## 4 EXPERIMENTS

We provide a broad empirical comparison of our proposed method in both continuous supervised and reinforcement learning settings. We first evaluate our method on synthetic data sets with time-varying shifts. Next, we create continuous supervised image classification tasks utilizing CIFAR-10 (Krizhevsky & Hinton, 2009) and CLEAR (Lin et al., 2021) datasets. Finally, we evaluate our method on ROBEL D'Claw (Yang et al., 2021) simulated robotic environments and MuJoCo (Todorov et al., 2012) simulator. See also Appendices C and D for a description of our settings and more results.

### 4.1 CONTINUOUS SUPERVISED LEARNING

The goal in these experiments is to build a model that can predict well on data from $t+1$ using *only* historical data from $\{p_0, \ldots, p_t\}$ (see also Remark 2). Note we train all different models, including all baselines, completely from *scratch* per time step; we do *not* update the model continually because we focus on understanding the performance of the time-varying importance weight only. Moreover, our method automatically detects shifts in the distribution of data without knowing them as a priori. In the following, we first explain the baseline models, then the data sets and simulated shift setups. Finally, we discuss the results of these experiments.

### 4.1.1 BASELINE METHODS

We compare our approach to the following baseline methods:

**1. Everything:** To obtain model used at time $t$, we train on pooled data from all past times $\{p_s(x) : s \leq t\}$, including the most recent data from $p_t(x)$. More precisely, this involves minimizing the objective

$$\ell^t(\phi) = \frac{1}{N} \sum_{s=1}^{t} \sum_{\{x_i : t_i = s\}} \ell(\phi; x_i). \tag{12}$$

where the loss of the predictions of a model parameterized by weights $\phi$ is $\ell(\phi; x_i)$ and we have defined $\ell^t(\phi)$ as the cumulative loss of all tasks up to time $t$.

**2. Recent** trains only on the most recent data, i.e., data from $p_t(x)$. This involves optimizing

$$\ell_t(\phi) = \frac{1}{N'} \sum_{\{x_i : t_i = t\}} \ell(\phi; x_i), \text{ where } N' = \sum_{\{x_i : t_i = t\}} 1. \tag{13}$$

**3. Fine-tune:** first trains on all data from the past using the objective

$$\phi^{t-} = \arg\min_{\phi} \frac{1}{N} \sum_{s=1}^{t} \sum_{\{x_i : t_i = s, s < t\}} \ell(\phi; x_i)$$

and then finetunes this model using data from $p_t$. We can write this problem as

$$\ell^{t+}(\phi) = \frac{1}{N'} \sum_{\{x_i : t_i = t\}} \ell(\phi; x_i) + \Omega(\phi - \phi^{t-}), \tag{14}$$

where $\Omega(\phi - \phi^{t-})$ is a penalty that keeps $\phi$ close to $\phi^{t-}$, e.g., $\Omega(\phi - \phi^{t-}) = \|\phi - \phi^{t-}\|_2^2$. Sometimes this approach is called "standard online learning" (Finn et al., 2019) and it resembles (but is not the same as) Follow-The-Leader (FTL) (Shalev-Shwartz, 2012). We call it "Finetune" make the objective explicit to avoid any confusion due its subtle differences with FTL. In practice, we implement finetuning without the penalty $\Omega$ by initializing the weights of the model to $\phi^{t-}$ in (14).

**Remark 4 (Properties of the different baselines).** Recall that our goal is to build a model that predicts on data from $p_{t+1}(x)$. It is important to observe that **Everything** minimizes the objective over all the past data; for data that evolves over time, doing so may be detrimental to performance at time $t$. If we think using a bias-variance tradeoff, the variance of the predictions of **Everything** is expected to be smaller than the variance of predictions made by **Recent**, although the former is likely to have a larger bias. For example, if the data shifts rapidly, then we should expect **Everything** to have a large bias and potentially a worse accuracy than that of **Recent**. But if the amplitude of the variations in the data is small (and drift patterns repeat) then **Everything** is likely to benefit from the reduced variance and perform better than **Recent**. **Fine-tune** strikes a balance between the two methods and although the finetuning procedure is not always easy to implement optimally, e.g., the finetuning hyper-parameters can be different for different times. As our results illustrate neither of these approaches show consistent trend in the experiments and highly depend on a given scenario which can be problematic in practice, e.g. sometimes **Everything** performs better than both but other times **Fine-tune** is better (compare their performance in Fig. 1).

**Remark 5 (How does the objective change when we have supervised learning or reinforcement learning?).** We have written Eqs. (12) to (14) using the loss $\ell(\phi; x_i)$ only for the sake of clarity. In general, the loss depends upon both the inputs $x_i$ and their corresponding ground-truth labels $y_i$ for classification problems. For problems in reinforcement learning, we can think of our data $x_i$ as entire trajectories from the system and thereby the loss $\ell(\phi; x_i)$ as the standard 1-step temporal difference (TD) error which depends on the weights that parameterize the value function.

We incorporate our time-varying importance weight into (12) as follows:

$$\ell^t(\phi) = \frac{1}{N} \sum_{s=1}^{t} \sum_{\{x_i : t_i = s\}} \boldsymbol{\omega}(x_i, t, t_i) \, \ell(\phi; x_i). \tag{15}$$

Note that the *only* difference between our method and **Everything** is introduction of $\boldsymbol{\omega}$ into (12) and all other details (e.g. network architecture, etc.) are exactly the same. $\boldsymbol{\omega}$ can be trained based on our algorithm explained in §3.1.

### 4.1.2 DRIFTING GAUSSIAN DATA

In this experiment, we design continuous classification tasks using a periodic shift where samples are generated from a Gaussian distribution with time-varying means ($\mu_t$) and standard deviation of 1. In particular, we create a data set, $D^t = \{(x_i^t, y_i^t)\}_{i=1}^{N}$, at each time step $t$ where $x_i^t \sim N(\mu_t, 1)$, $\mu_t = \mu_{t-1} + d/10$, $\mu_0 = 0.5$, and label 1 ($y_i^t = 1$) is assigned if $x_i^t > \mu_t$ otherwise $y_i^t = 0$. We change direction of shift every 50 time steps (i.e., set $d \leftarrow -d$) and use $N = 2000$ in our experiments. We run this experiments for 160 time steps. Fig. 1a displays that our method performs much better than **Everything** and similarly to others. We note that both **Recent** and **Fine-tune** are like oracle for this experiment as test time data from $p_{t+1}(x)$ are more similar to the recent data than other historical data. This experiment makes a clear case for our method as it shows that our method allows the training to selectively utilize data similar to the current time and importantly data that evolved in a similar fashion in the past.

### 4.1.3 IMAGE CLASSIFICATION WITH EVOLVING LABELS

We adopt the setting of Wu et al. (2021) to design experiments for classification under continuous label shift with images from CIFAR-10 (Krizhevsky & Hinton, 2009) and CLEAR (Lin et al., 2021).

**CIFAR-10.** We split the original CIFAR-10 training data into 90% (train) and 10% (validation), and use original test split as is. Following Wu et al. (2021), we pre-train Resent-18 (He et al., 2016) using training data for 100 epochs. In subsequent continuous label shift experiments, we utilize the pre-trained model and only fine-tune last convolution block and classification layer. We average all results over 3 random splits.

**CLEAR.** This data set consists of 10 classes where each class has 3000 samples. CLEAR is curated over a large public data set only including data spanning from years 2004 to 2014. We utilize unsupervised pre-trained features provided by Lin et al. (2021) and only use a linear layer classifier As Lin et al. (2021) stated, the linear model using pre-trained features is far more effective than training from scratch for this data set (For more, see Lin et al. (2021)). We average all results over 3 random train-validation-test splits, of relative size 60% (train), 10% (validation), and 30% (test).

**Continuous label shifts.** We create a sequence of classification tasks where the label distribution changes at each time according to a predefined label distribution in slow repeating patterns (see Appendix C.1 for details). In particular, we train a model at time $t$ from scratch using all training data seen so far and evaluate the model on test data drawn from label distribution of time $t + 1$. In this setting, training data keeps growing while test data stays the same size. This is a challenging problem as label shift occurs at each time step hence it is critical for a method to learn how to only utilize data which are relevant for the current time step and task. Note we only use test data in the evaluation time and never use it to train a model.

**Results.** We report results comparing our method against the baselines across different data sets and shifts in Fig. 1. These results clearly illustrate that our method performs better than others as our method obtains higher classification accuracy across all benchmarks. Importantly, our method shows consistent performances across various shifts and benchmarks whereas other methods do not. For instance, **Everything** fares poorly in Gaussian experiment than others but it works better than **Fine-tune** and **Recent** on continuous CIFAR-10 benchmark. We also shows in Fig. 2 that our method produces high quality estimates under shifted distribution.

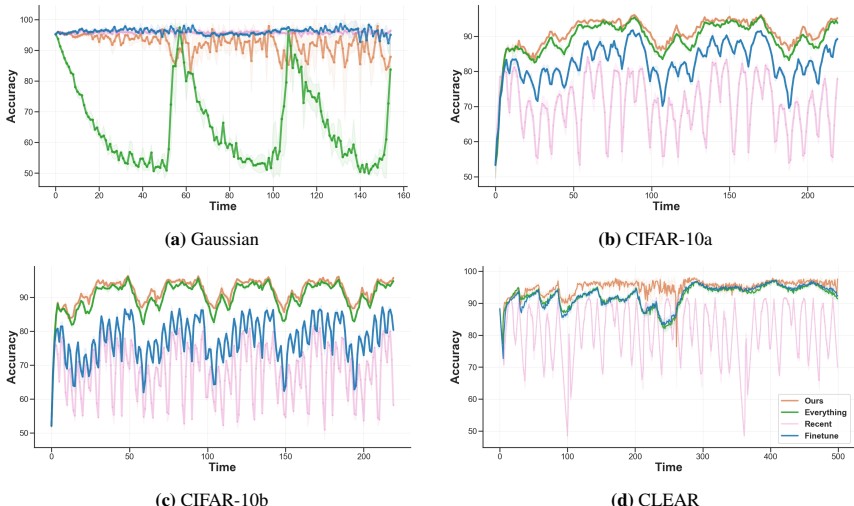

**Figure 1: Average test accuracy (higher is better) across 3 runs for continuous supervised learning benchmarks comparing our method with others**. The x-axes indicate different test time steps $t$ and the y-axes display the classification accuracy on test data from time $t$. CIFAR-10a and CIFAR-10b refer two different shifts for this benchmark (see Appendix C.1 for details). All models in these experiments are trained completely from scratch per time step (i.e. every point in these plots is a different model) using exact same architecture and hyper-parameters. A sudden drop in a plot is an indication that the shift direction has changed. We see that our method is the only approach that consistently works across different settings and shifts.

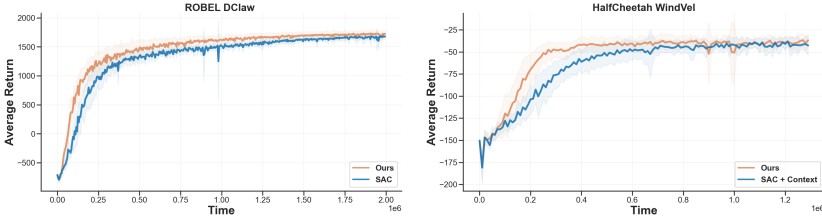

**Figure 3: Comparison of the average undiscounted return (higher is better) of our method (orange) against baseline algorithms on ROBEL D'Claw and Half-Cheetah WindVel environments**. In all environments, our method performs better than other methods in terms of sample complexity.

## 4.2 REINFORCEMENT LEARNING WITH CHANGING TASKS

We use two different environments for experiments in this section. For all experiments in this section, we report the undiscounted return averaged across 10 different random seeds.

**ROBEL D'Claw.** This is a robotic simulated manipulation environment containing 10 different valve rotation tasks and the goal is to turn the valve below 180 degrees in each task. All tasks have the same state and action spaces, and same episode length but each rotates a valve with a different shape. We build a random sequence of these tasks switching from one to another (after an episode finishes) in repeating patterns. We utilize SAC (Haarnoja et al., 2018) as our baseline. In order to incorporate our approach into

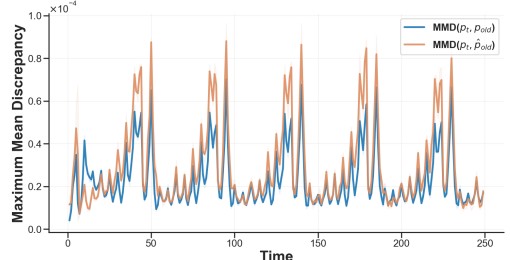

**Figure 2: Maximum Mean Discrepancy (MMD) of true distributions $(p_t, p_{\text{old}})$ and $(p_t, \hat{p}_{\text{old}})$ where $\hat{p}_{\text{old}}$ is constructed from past data weighted by our method**. Each (blue) point in this plot shows distance between data from $p_t(x)$ and all data from $p_{\text{old}} := \{p_{t_i(x)}\}_{t_i < t}$. Estimates for $\hat{p}_{\text{old}}$ are based on 200 images sampled at each time step from a drifting version of CIFAR-10. As the underlying distributions evolve, their MMD distance grows. To illustrate that our method produces high quality estimates, we also show distance between data from $p_t$ and all data from $\{p_{t_i}(x)\}_{t_i < t}$ which are *weighted* by our $\omega$. The small gap between blue and orange plots shows that our method produces accurate estimates.

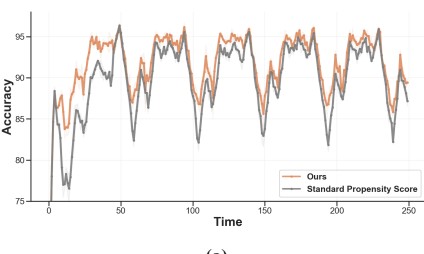 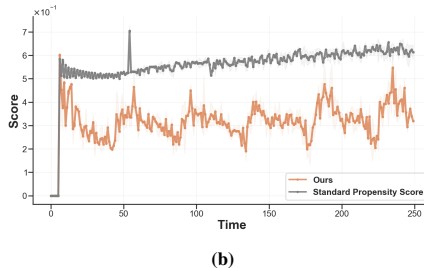

(a)                                                (b)

**Figure 4: Comparing our method against standard propensity on continuous CIFAR-10 benchmark**. Fig 4a shows average test accuracy and Fig 4b shows propensity scores for both methods across time during test time. These results distinctly show our time-varying importance weight estimator weighs data in the same fashion as they evolve and this can explain why our method performs better than standard propensity score where it can not identify when data evolves.

SAC, we only change Q-value update with $\boldsymbol{\omega}$ as follows:

$$\frac{1}{|B|} \sum_{(s,a,t,s')} \left[ \boldsymbol{\omega}(s,a,T,t) \Big( Q_\psi(s,a) - r - \gamma Q_{\hat{\psi}}(s',a') + \alpha \log \pi_\phi(a'|s') \Big)^2 \right], a' \sim \pi_\phi(\cdot|s') \quad (16)$$

where $\alpha$ is entropy coefficient, $T$ is the current episode, and $(s,a)$ denote state and action, respectively.

**Half-Cheetah WindVel.** We closely follow the setup of Xie et al. (2020) to create *Half-Cheetah WindVel* environment where direction and magnitude of wind forces on an agent keep changing but slowly and gradually. This is a difficult problem as an agent needs to learn a non-stationary sequence of tasks (i.e. whenever direction and magnitude of wind changes, it constitutes as a new task). Since task information is not directly available to the agent, we utilize the recent method of Caccia et al. (2022) and Fakoor et al. (2020b) to learn and infer task related information using context variable (called SAC + Context here). Similar to (16), we only change Q-update to add our method.

**Results.** We can see in Fig. 3 that our method offers consistent improvements over the baseline method and can observe its effectiveness in correcting shifts across time. Note that the only difference between our method and baseline methods is the introduction of $\boldsymbol{\omega}$ term in the Q-update and all other details are exactly the same. See also Fig. 6 and 7 in appendix for additional experiments.

## 5 COMPARING STANDARD PROPENSITY WITH OUR TIME-VARYING METHOD

Now we compare our method with standard propensity scoring discussed in §3. Fig. 4 shows results of this experiment on CIFAR-10 benchmark. This result provides an empirical verification that our method works better than standard propensity method and gives us some insights why it performs better. Particularly, we can see in Fig. 4 that our method fairly detects shifts across different time steps, whereas standard propensity scoring largely ignores shifts across different time steps.

## 6 CONCLUSION

Here we propose a simple yet effective time-varying importance weight estimator to account for gradual shifts in data distributions. One practical property of our drift estimator is that it is not specific to a particular class of problems and can be utilized in all kinds of settings from supervised learning to reinforcement learning with a combination of existing approaches. Importantly, our method automatically detects shift in data without knowing them as a priori. Comprehensive experiments on continuous supervised learning and reinforcement learning show the effectiveness of our method. The broader impact of this work will potentially be to make machine learning methods more robust in the presence of continuous distribution shift, which is inevitable in the real world. Note that while we referred to $t$ as time throughout this paper, our methodology is also applicable to settings where data-generating shifts occur gradually over space or any other continuous-valued index.

## 7  ETHICS STATEMENT

All datasets and benchmarks used in this work are publicly available. All code and documentation developed as a part of this project will made public. We do not declare any conflicts of interest with respect to the findings discussed in this work.

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

# A  DERIVATIONS

Using a similar calculation as that of (1), we can write our time-varying importance estimator objective as follows

$$\mathbb{E}_{x\sim p_T(x)}\Big[\ell(x)\Big] = \mathbb{E}_{x\sim p(x|T)}\Big[\ell(x)\Big] = \int_t dp(x|T)\,\ell(x) = \int_t\int_x dp(t)\,dp(x|T)\,\frac{dp(x|t)}{dp(x|t)}\ell(x) \quad (17)$$

$$= \int_t\int_x dp(t)\,dp(x|t)\,\frac{dp(x|T)}{dp(x|t)}\ell(x)$$

$$= \int_t dp(t)\int_x dp(x|t)\,\frac{dp(x|T)}{dp(x|t)}\ell(x)$$

$$= \mathbb{E}_{t\sim p(t)}\mathbb{E}_{x\sim p(x|t)}\Big[\frac{dp(x|T)}{dp(x|t)}\ell(x)\Big]$$

$$= \mathbb{E}_{t\sim p(t)}\mathbb{E}_{x\sim p_t(x)}\Big[\frac{dp_T(x)}{dp_t(x)}\ell(x)\Big]$$

$$= \mathbb{E}_{t\sim p(t)}\mathbb{E}_{x\sim p_t(x)}\Big[\omega(x,T,t)\ell(x)\Big] \quad (18)$$

# B  ALGORITHM DETAILS

Algorithm 1 describes the detailed implementation steps of our time-varying importance weight estimator. We use the exact same steps for both reinforcement learning and continuous supervised learning experiments. The main difference between these cases is using of $(x,y,t)$ as inputs for supervised learning tasks whereas $(s,a,t)$ are used as inputs in the reinforcement learning experiments. Note although $g_\theta$ can be parameterized differently considering its given task (e.g. image classification tasks require $g_\theta$ to be a convolutional deep network; however, continuous control tasks in reinforcement learning experiments need fully connected networks), its implementation details remain the same.

In order to implement Method 2 described in §3, we only need to change line 8 of Algorithm 1 as follow (all other details remain the same):

$$\nabla_\theta J \leftarrow \nabla_\theta \sum_{(x,t_2,z)\in\mathcal{B}} \log\Big(1 + e^{-z(g_\theta(x,t_2))}\Big) \quad (19)$$

For all the experiments in this paper, we utilize 19 in Algorithm 1 implementation. Fig. 5 compares these methods using the continuous supervised learning benchmarks and shows that both method performs similarly.

---

**Algorithm 1** Train Time-varying Importance Weight Estimator

---

1: **Input**: $\mathcal{D} = \{x_j, t_j\}_{j=1}^N$
2: **Input** $M$: Number of epochs
3: Initialize $\mathcal{D}_o = \varnothing$
4: **for** j=1...M **do**
5: $\quad \mathcal{D}_o \leftarrow GenerateData(D)$
6: $\quad$ **repeat**
7: $\quad\quad$ Sample mini-batch $\mathcal{B} = \{(x, t_2, t_1, z)\} \sim \mathcal{D}_o$
8: $\quad\quad \nabla_\theta J \leftarrow \nabla_\theta \sum_{(x,t_2,t_1,z)\in\mathcal{B}} \log\Big(1 + e^{-z(g_\theta(x,t_2)-g_\theta(x,t_1))}\Big)$
9: $\quad\quad \theta \leftarrow \theta - \alpha\nabla_\theta J$
10: $\quad$ **until** convergence
11: **end for**

---

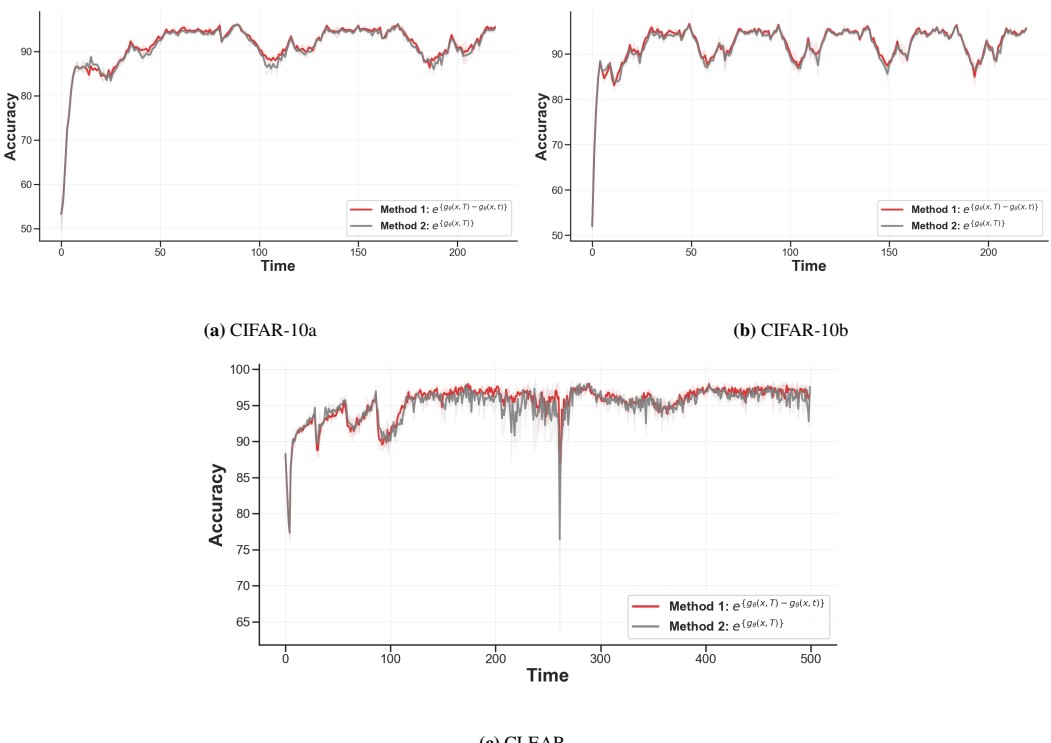

**(a)** CIFAR-10a

**(b)** CIFAR-10b

**(c)** CLEAR

**Figure 5: Comparing our Method 1 (see (8)) and Method 2 (see (10)) on continuous CIFAR-10 and CLEAR benchmarks**. These results shows average test accuracy across time during test time. Both methods use exactly the same setting and the only difference between them is utilizing Method 1 or Method 2 for time-varying importance weight estimator. These results show that both methods perform similarly, although Method 1 has a slight edge over Method 2.

---

**Algorithm 2** $GenerateData$

---

1: **Input**: $\mathcal{D} = \{x_j, t_j\}_{j=1}^{N}$
2: $\mathcal{T} = [1, .., T]$
3: Initialize $\mathcal{D}_o = \varnothing$
4: **for** j=1...N **do**
5:    $t_r$ is chosen uniformly at random from $\mathcal{T}$ such that $t_j \neq t_r$
6:    **if** random() $>= 0.5$ **then**
7:       $z = 1$
8:       $\mathcal{D}_o \leftarrow \mathcal{D}_o \cup \{(x_j, \mathbf{t_j}, t_r, z)\}$
9:    **else**
10:      $z = -1$
11:      $\mathcal{D}_o \leftarrow \mathcal{D}_o \cup \{(x_j, t_r, \mathbf{t_j}, z)\}$
12:    **end if**
13: **end for**
14: **Return** $\mathcal{D}_o$

---

## C  EXPERIMENT DETAILS

**Implementation Details.** Table 3, 2, and Table 1 show hyper-parameters, computing infrastructure, and libraries used for the experiments in this paper. We did minimal random hyper-parameters search for the experiments in this paper and we mostly follow standard and available settings for the experiments whenever it is applicable. We will open source the code of this paper upon publication.

| Computing Infrastructure | |
|---|---|
| Machine Type | AWS EC2 - g3.16xlarge |
| CUDA Version | 11.0 |
| NVIDIA-Driver | 450.142.00 |
| GPU Family | Tesla M60 |
| CPU Family | Intel Xeon 2.30GHz |
| **Library Version** | |
| Python | 3.8.5 |
| Numpy | 1.22.0 |
| Pytorch | 1.10.0 |

**Table 1:** Software libraries and machines used in this paper.

### C.1 CONTINUOUS LABEL SHIFT SETTINGS

To simulate the continuous label shift process for image classification experiments in this paper, we follow the setting of Wu et al. (2021) but adopt their settings to more challenging scenarios. In particular, we simulate label shifts for all classes (i.e. 10 classes here) as opposed to two classes in their setting:

$$\forall i \in [1, C] \ \ \forall t \in [1, \mathcal{T}], v_t = (1 - \frac{t}{\mathcal{T}})q_t^i + (\frac{t}{H})q_t^{i+1}, \tag{20}$$

where $\mathcal{T}$ is the number of shift steps per a label pair $i$ and $i + 1$, $t \in [1, \mathcal{T}]$, $C$ denotes the number of classes ($C = 10$ in our experiments), $q_t^i \in R^C$ is vector of class probabilities, and $v_t \in R^{\mathcal{T} \times C}$. For CIFAR-10 experiments, we use $\mathcal{T} = 9$ (called CIFAR-10a), $\mathcal{T} = 6$ (CIFAR-10b), and $\mathcal{T} = 30$ (CIFAR-10c). We also use $\mathcal{T} = 30$ for CLEAR experiment. Sample python implementation of (20) is shown in Listing 1.

```python
import numpy as np
def create_shift(T, q1, q2):
    lamb = 1.0 / (T-1)
    return np.concatenate([np.expand_dims(q1 * (1 - lamb * t) + q2 * lamb
      * t, axis=0) for t in range(T)], axis=0)
```

Listing 1: **Python code of (20) for given two classes**

## D MORE RESULTS

**Number of seeds.** We repeat some of supervised learning experiments of CIFAR-10 benchmark using 8 different random seeds (we increase the number of seeds from 3 to 8) and report results in Fig. 9. As these new experiments show, results are consistent across 3 and 8 seeds. We also like to emphasize that all our reinfrocement leatning experiments are done for 10 seeds which is standard for MuJoCo environments.

**Method 1 vs Method 2.** We compare performance of our Method 1 (8) and Method 2 (10) on continuous CIFAR-10 and CLEAR benchmarks. We can see in Fig. 5 that both methods perform similarly, although Method 1 has a slight edge over Method 2. Note we use Method 2 for all the experiments in the paper.

**No Shift.** Here, we analyze how our method performs when there is no shift in data and see whether or not it performs as good as **Everything**? To answer this question, we run new experiments on the CIFAR-10 benchmark *without* shift. We build continuous classification tasks from this dataset where we used past data to predict future data points. This is the same setting as other supervised learning experiments in the paper except we do not apply any shift to the data. Results of this experiment are shown in Fig. 10. As this experiment shows, our method and **Everything** perform similarly when

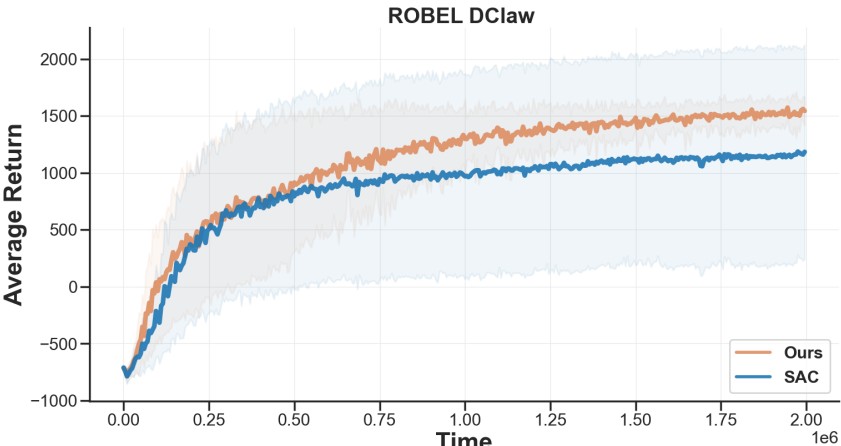

**Figure 6: Comparison of the average undiscounted return (higher is better) of our method (orange) against baseline algorithm on ROBEL D'Claw**. In this environment, our method performs better than standard SAC in terms of both sample complexity and stability (compare shaded area of our method with SAC, larger means less stable). We can see that our method has much more stable performance than SAC. Note the only difference between this experiment and the one in Fig. 3 is that here we only move to next task after repeating each task for 215 time steps whereas each task repeats only for 2 time in the experiment of Fig. 3. Our method works better in both settings and this result again verifies the effectiveness of our method in practice.

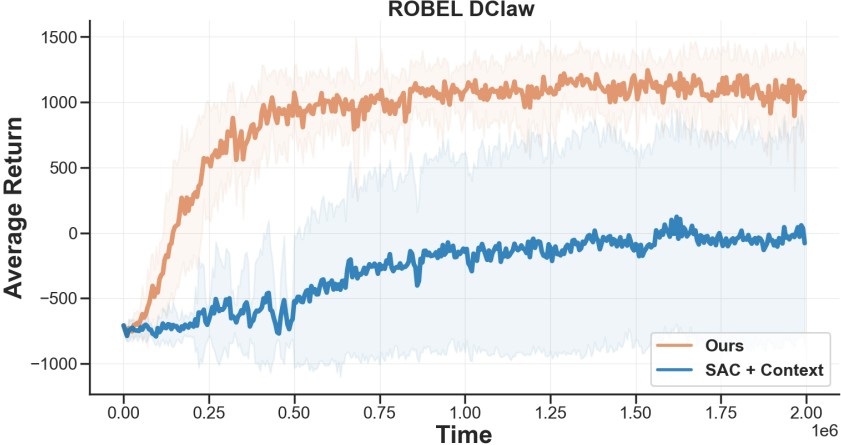

**Figure 7: Comparison of the average undiscounted return (higher is better) of our method with context (orange) against baseline algorithm with context on ROBEL D'Claw**. In this environment, our method performs better than SAC + Context in terms of both sample complexity and stability (compare shaded area of our method with SAC + Context, larger means less stable). Note that the only difference between our method and baseline method is the introduction of $\omega$ term in the Q-update and all other details are *exactly* the same.

| Hyper-parameters for Algorithm 1 | value |
|---|---|
| Learning rate | 1e-4 |
| Number of Updates (M) | 4 |
| Batch Size | 512 |
| $g_\theta$ | 4-FC layers with *ReLU* and *BatchNorm* |
| Hyper-parameters of SAC | value |
| Random Seeds | $\{0, 1, .., 9\}$ |
| Batch Size | 256 |
| Number of $Q$ Functions | 2 |
| Number of hidden layers (Q) | 2 layers |
| Number of hidden layers ($\pi$) | 2 layers |
| Number of hidden units per layer | 256 |
| Nonlinearity | *ReLU* |
| Discount factor ($\gamma$) | 0.99 |
| Target network ($\psi'$) update rate ($\tau$) | 0.005 |
| Actor learning rate | 3e-4 |
| Critic learning rate | 3e-4 |
| Optimizer | Adam |
| Replay Buffer size | 1e6 |
| Burn-in period | 1e4 |
| $\omega$ clip | 1 |
| HalfCheetah-Wind's number of episodes to evaluate | 50 |
| Robel's number of episodes to evaluate | 5 |

**Table 2:** Hyper-parameters used for all reinforcement learning experiments. All results reported in our paper are averages over repeated runs initialized with *each* of the random seeds listed above and run for the listed number of episodes.

| Hyper-parameters for Algorithm 1 | value |
|---|---|
| Learning rate | 1e-4 |
| Number of Updates (M) | 200 |
| Batch Size | 512 |
| $g_\theta$ for CIFAR-10 | 5-Conv layers and 1 linear layer |
| $g_\theta$ for CLEAR | 3-FC layers with *ReLU* and *BatchNorm* |
| Hyper-parameters of CIFAR-10/CLEAR | value |
| Random Seeds | $\{0, 1, 2\}$ |
| CIFAR-10 Network | Resnet-18 |
| CLEAR Network | Linear layer |
| Samples per time step | 200 |
| Learning rate | 9e-4 |
| Batch Size | 128 |
| Optimizer | Adam |
| $\omega$ clip | 1 |
| Number of training epochs per time step | CIFAR-10 (20), CLEAR (25) |

**Table 3:** Hyper-parameters used for all continuous supervised learning experiments for both CIFAR-10 and CLEAR tasks. All continuous supervised learning results reported in our paper are averages over three random seeds.

there is no shift in the data. This experiment provides further evidence about the applicability of our method and shows that our method works regardless of presence of shifts in the data and it has no negative effect on the performance.

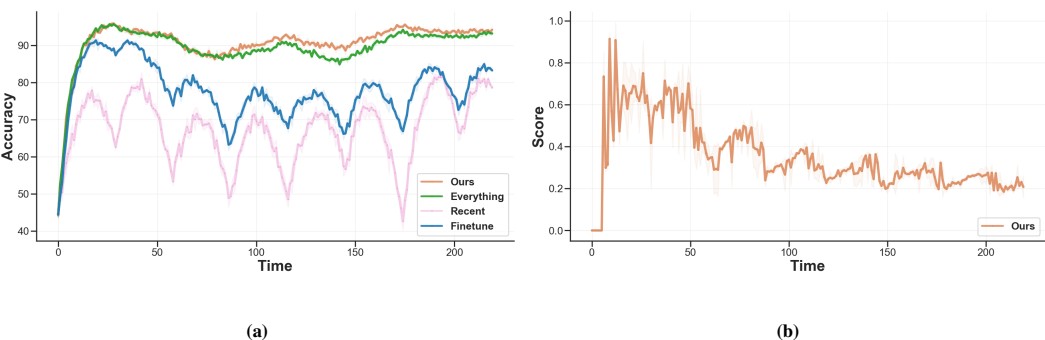

(a)  (b)

**Figure 8: Average test accuracy (higher is better) across 3 runs for continuous supervised learning on CIFAR-10c (see Appendix C.1 for details) benchmark comparing our method with others**. The x-axes indicate different test time steps. The y-axis on the left displays the classification accuracy on test data and the one on the right shows our time-varying importance weight scores. Note all models in these experiments are trained completely from scratch per time step (i.e. every point in this plot is a different model) using exact same architectures and hyper-parameters. A sudden drop in a plot is an indication that the shift direction has changed. We see that our method is the only method showing consistent behavior across different benchmarks and shifts, and importantly performs the best overall.

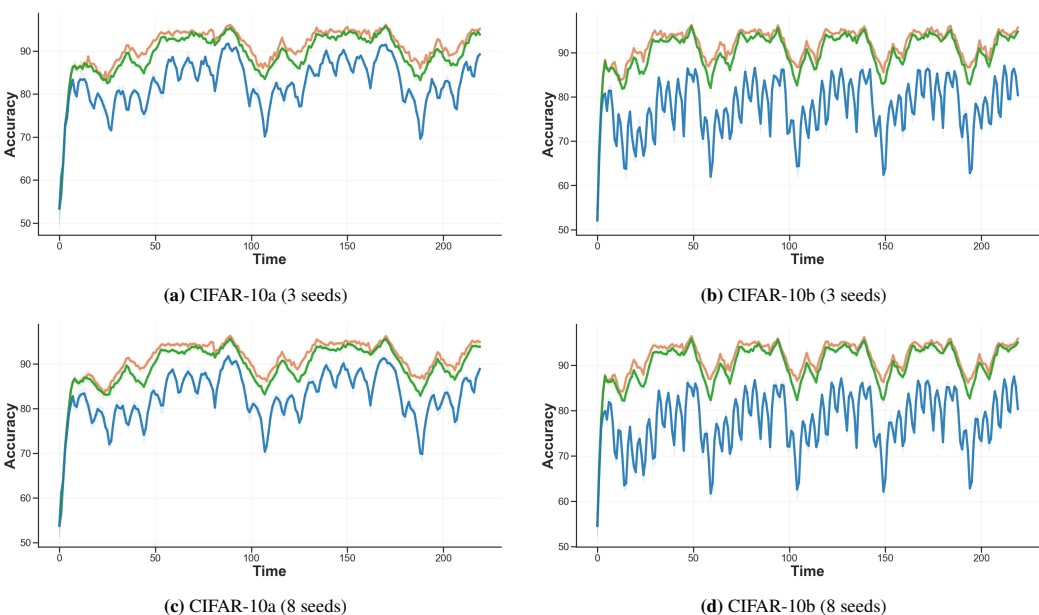

**Figure 9: Comparison of average test accuracy (higher is better) of running with 3 seeds against 8 seeds on continuous supervised learning benchmarks**. The x-axes indicate different test time steps $t$ and the y-axes display the classification accuracy on test data from time $t$. CIFAR-10a and CIFAR-10b refer two different shifts for this benchmark (see Appendix C.1 for details). We see that results are consistent across 3 seeds and 8 seeds.

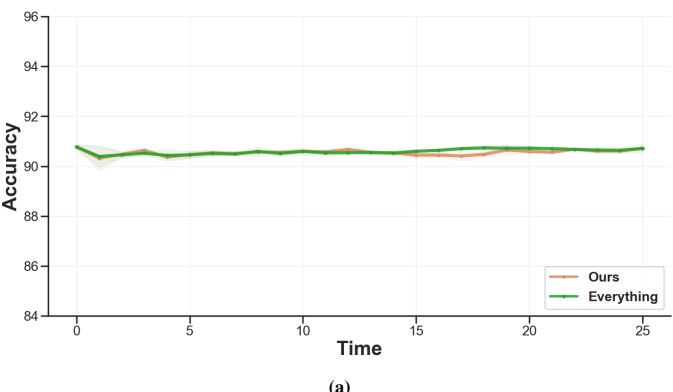

(a)

**Figure 10: Average test accuracy (higher is better) across 3 runs for continuous supervised learning on CIFAR-10 benchmark *without* label shift** . The x-axis indicates different test time steps and the y-axis displays the classification accuracy on test data. We see that our method and **Everything** perform similarly. This is an important observation showing our method does not negatively affect the performance when there is no shift.

