# OpenReview forum: "Data Drift Correction via Time-varying Importance Weight Estimator"
_ICLR.cc/2023/Conference — Submitted to ICLR 2023_

### Official Review · Reviewer_TuzH · 2022-10-22

**Confidence:** 3
**Correctness:** 3
**Technical Novelty And Significance:** 3
**Empirical Novelty And Significance:** 2
**Recommendation:** 5

**Clarity, Quality, Novelty And Reproducibility:**

The paper is generally well-written, but there are some points that may deserve further attention.

- Analyzing the algorithm details reported in the appendix, I found some aspects that are not clear to me. It seems there is a mistake in the definition of Algorithm2 "Generate Data" as the generation of the tuple in the case of z=-1 leads to triplets of the form (x_i, t', t) where t_i does not correspond to t, as instead stated in the definition of Method 1. It seems that the provided formulation of the function "Generate Data" is someway more appropriate to Method2. Also, in the section "Algorithm details" of the Appendix, the formulation (19) seems wrong as it never uses the value t_1 contained in the tuple. Could the authors try to clarify this point?
- Concerning the experimental section, it presents some critical aspects: the results reported for the Gaussian experiment do not show any advantage in using the proposed correction compared to the Recent and Finetune baselines. It is also not clear how, in the experiments with CIFAR-10, the Finetune approach shows worse performances than the Everything approach. Do you have an intuitive explanation for this? Furthermore, the authors only perform 3 runs for this set of experiments. I would suggest to report results averaged over more runs, as done for the Reinforcement Learning's experiments.
- Regarding the experiments related to the Reinforcement Learning, would it have been more appropriate to use as baselines other algorithms suited for non-stationary settings, besides only using vanilla SAC as done in ROBEL D'Claw environment?
- Another aspect is that, in the main paper, the authors claim that the experiments will show the performance of both method 1 and 2, however, in the appendix they state that for the experiments they only used method 2, thus not providing experiments using method 1. I would suggest the authors to also show the results of method 1.

The approach of reweighting past samples in gradually drifting environments seems also novel in the literature. As for the reproducibility, the authors report in the appendix all the hyperparameters needed to run the different experiments, but they do not provide the source code.

**Details Of Ethics Concerns:**

None.

**Strength And Weaknesses:**

**Strengths**
- The approach is simple to understand
- The approach is applicable to different settings from supervised to reinforcement learning

**Weaknesses**
- The approach seems computational (and possibly sample) demanding due to the need for retraining the binary classifier (for the weights) and of the prediction model at each time step.
- The experimental evaluation does not succeed in showing a clear advantage of the proposed method

**Summary Of The Paper:**

The paper deals with handling changes in non-stationary environments. The authors assume that data change gradually over time: this idea is formally made explicit using a constant to upper bound the total variation divergence of the distributions of data in a small amount of time. The assumption thus allows predicting samples at time T+dT (in the future) using a time-dependent propensity score built on samples at time T. The authors try to reuse past samples (t<=T) in order to make predictions on samples at time T by using a mechanism based on importance weighting. The weights are computed using the output of a probabilistic binary classifier that maps each sample to the time it was seen, that coincides with a specific distribution of data. In the end, the authors test their approach in both continuous supervised and reinforcement learning settings comparing it with different baselines.

**Summary Of The Review:**

The paper tackles the important problem of non-stationarity of data using the standard technique of importance weighting enriched with a time component in order to help the mapping of each sample to its associated time of arrival. The main ideas and assumptions presented in the paper are sound. The use of an importance-weighted estimator provides an unbiased estimate of the samples arriving at time T, which is a good proxy for the samples to come in the near future. Nevertheless, the concerns raised make me opt for a borderline negative score at the moment.

---

> ### Author Response · Authors · 2022-11-18
> **Response to Reviewer TuzH -- Part 1**
>
> Thank you for your feedback and suggested experiments. Please also see the main comment above. We hope you will consider increasing your score after seeing our response.
>
> > The approach seems computational (and possibly sample) demanding due to the need for retraining the binary classifier (for the weights) and of the prediction model at each time step.
>
> We simply fit a nonlinear logistic regression to estimate the importance weight (as we show in the Algorithm 1). The computational complexity of estimating the importance weight is marginal as compared to training the deep network for the main task.
>
> > Analyzing the algorithm details reported in the appendix, I found some aspects that are not clear to me. It seems there is a mistake in the definition of Algorithm2 "Generate Data" as the generation of the tuple in the case of z=-1 leads to triplets of the form (x_i, t', t) where t_i does not correspond to t, as instead stated in the definition of Method 1. It seems that the provided formulation of the function "Generate Data" is someway more appropriate to Method2. Also, in the section "Algorithm details" of the Appendix, the formulation (19) seems wrong as it never uses the value t_1 contained in the tuple. Could the authors try to clarify this point?
>
> Thank you. You are right, there were a typo in the Algorithm2 "Generate Data". We now updated the  Algorithm2 "Generate Data" in the revised paper.
>
> > Concerning the experimental section, it presents some critical aspects: the results reported for the Gaussian experiment do not show any advantage in using the proposed correction compared to the Recent and Finetune baselines.
>
> Great observation. In fact that is the point of these experiments: our method shows **consistent** behavior across various shifts, settings, and benchmarks whereas other methods do not have consistent performance and are sensitive to a given shift and setting. In particular, Everything fares poorly in Gaussian experiments than others but it works better than Fine-tune and Recent on continuous CIFAR-10 benchmark.
>
> > It is also not clear how, in the experiments with CIFAR-10, the Finetune approach shows worse performances than the Everything approach. Do you have an intuitive explanation for this?
>
> As we discussed in the paper, although Fine-tune strikes a balance between Recent and Everything methods, the finetuning procedure is not always easy to implement optimally, e.g., the finetuning hyper-parameters can be different for different times. As our results illustrate, none among Fine-tune, Everything, and Recent, show a consistent trend in the experiments and whether they work well depends highly on a given scenario; this can be problematic in practice.
>
> > the authors only perform 3 runs for this set of experiments. I would suggest to report results averaged over more runs, as done for the Reinforcement Learning's experiments.
>
> Per the reviewer’s request, we repeated our supervised experiments of CIFAR-10a/b for our method, Everything, and Fine-tune for 8 different seeds (we increased the number of seeds from 3 to 8) and reported results in Figure 9. As these new experiments show, our results are consistent across 3 and 8 seeds. We also like to emphasize that all our RL experiments are done for 10 seeds which is standard for MuJoCo environments.
>
> > Regarding the experiments related to the Reinforcement Learning, would it have been more appropriate to use as baselines other algorithms suited for non-stationary settings, besides only using vanilla SAC as done in ROBEL D'Claw environment?
>
> That is a great question. We clarify that using context variable with an off-policy method ( (i.e. called SAC + Context here) is a strong method for non-stationary settings in RL (Caccia et al., 2022, Ni et al., 2022, and Fakoor et al., 2020) and leads to state-of-art performance in those settings, and it is utilized for the non-stationary settings experiments in this work. We built context using RNN as discussed in Caccia et al., 2022, Ni et al., 2022, and Fakoor et al., 2020.
>
> Per reviewer request,  we ran new experiments and used SAC + Context for ROBEL D'Claw. Figure 7 shows the new results. In all experiments, our methods show a clear advantage over baseline methods. In addition, Figure 3 and Figure 5 demonstrate results when we incorporate our method with standard SAC in different ROBEL D'Claw settings. These results further highlight the applicability and flexibility of our approach when it incorporates different methods.
>
> [Caccia et al., 2022] Task-agnostic continual reinforcement learning: In praise of a simple baseline
>
> [Fakoor et al., 2020] Meta-q-learning.
>
> [Ni et al., 2022]  Recurrent Model-Free RL Can Be a Strong Baseline for Many POMDPs

---

> > ### Author Response · Authors · 2022-11-18
> > **Response to Reviewer TuzH -- Part 2**
> >
> > > the authors claim that the experiments will show the performance of both method 1 and 2, however, in the appendix they state that for the experiments they only used method 2, thus not providing experiments using method 1. I would suggest the authors to also show the results of method 1
> >
> > Thank you for your suggestion. We only reported results for method 2 in the first draft of the paper. However, per your suggestion, we have now included new results for method 1 (and compared with method 2) in Figure 5 of the revised paper. These results show that these two methods perform similarity while method 1 has a slight edge over method 2. Note that all results in the paper are still based on Method 2 and these new results show that it doesn’t matter which method we use, we will get the same results and conclusions.
> >
> > > As for the reproducibility, the authors report in the appendix all the hyperparameters needed to run the different experiments, but they do not provide the source code.
> >
> > Good point. We will release the source code of the paper upon publication.

---

### Official Review · Reviewer_UiAM · 2022-10-25

**Confidence:** 3
**Clarity, Quality, Novelty And Reproducibility:** The paper is of good quality, and ver…
**Correctness:** 4
**Technical Novelty And Significance:** 3
**Empirical Novelty And Significance:** 3
**Recommendation:** 6

**Strength And Weaknesses:**

The method is very well described, and the results on synthetic data are very compelling.

Weaknesses:

I find that the "CONTINUOUS LABEL SHIFT SETTINGS" paragraph in the appendix is not detailed enough. It is hard to get a sense of the distribution shift that is induced in the data. Maybe adding some visualization here would be great.
Also, there is a public "corrupted" CIFAR10 dataset that allows to get CIFAR10 data, with different levels of corruptions. It might be useful (and more standard) to use this dataset for further analysis.

An important component of supervised learning models is their uncertainty representation. if the proposed method correctly approximates the distribution shift, it should be reflected in the uncertainty representation and in the calibration of the model. How well does the model perform in terms of ECE, for example, and how does the uncertainty evolves as distribution shifts? I think such analysis would strengthen the paper.

Finally, I would have liked to see real data experiments. There are some "common" settings and datasets for distributional shifts. For example, in neuroscience, decoding neural data during learning would be one (the distribution of the input changes as the animal learns the task) or in climate science (the basic models do not accurately forecast weather or temperature as the CO2 emissions increase, for example)



**Summary Of The Paper:**

The authors present a novel time-varying importance weight estimator that can detect gradual shifts in the distribution of data. They evaluate their methods on synthetic image classification and reinforcement learning tasks that exhibit distribution shifts and show that the proposed method largely outperforms the standard methods (training on all data, "recent" data only, and fine-tuning on the recent data).

**Summary Of The Review:**

The derivations are sound and the proposed model is very interesting, but the evaluation lacks a bit of depth and real-data experiments.

---

> ### Author Response · Authors · 2022-11-18
> **Response to Reviewer UiAM**
>
> Thank you for your feedback and suggested experiments. Please also see the main comment above. We hope you will consider increasing your score after seeing our response.
>
> > I find that the "CONTINUOUS LABEL SHIFT SETTINGS" paragraph in the appendix is not detailed enough. It is hard to get a sense of the distribution shift that is induced in the data. Maybe adding some visualization here would be great.
>
> We have updated the CONTINUOUS LABEL SHIFT SETTINGS paragraph in the appendix and provide a python code snippet to clarify how it is implemented.  Please see the revised paper. Hope this addresses your concerns.
>
> We will release the source code of the paper upon publication.
>
> > Also, there is a public "corrupted" CIFAR10 dataset that allows to get CIFAR10 data, with different levels of corruptions. It might be useful (and more standard) to use this dataset for further analysis.
>
> Thank you for the pointer. To create "corrupted" CIFAR10 dataset, Hendrycks 2019 apply diverse and different shifts to the data which lead to **abrupt** changes in the data distribution whereas we look into problems where data evolves **gradually**. For this kind of (abrupt and swift) distribution shift, one needs to build a method that has an adaptation step at test time (see Zhang et al. 2022, Wang  et al. 2021). Hence the setting of "corrupted" CIFAR10 is very different from the setting of our paper and it does not apply to the problem setting studied in our work.
>
> [Hendrycks,  2019 ] Benchmarking Neural Network Robustness to Common Corruptions and Perturbations
>
> [Zhang et al., 2022] MEMO: Test Time Robustness via Adaptation and Augmentation
>
> [ Wang et al., 2021] Tent: Fully Test-time Adaptation by Entropy Minimization
>
> > An important component of supervised learning models is their uncertainty representation. if the proposed method correctly approximates the distribution shift, it should be reflected in the uncertainty representation and in the calibration of the model. How well does the model perform in terms of ECE, for example, and how does the uncertainty evolves as distribution shifts? I think such analysis would strengthen the paper.
>
> We agree that uncertainty quantification and calibration of models is important. But we have focused on how to select past samples that are similar to the current distribution. Our method is designed to predict correctly, on average, on $p_{T+dt}$. It is entirely possible that these predictions are not well-calibrated---many other deep learning models. But calibration and uncertainty representation are not the focus of this paper, we may want to investigate them in future work.
>
> > Finally, I would have liked to see real data experiments. There are some "common" settings and datasets for distributional shifts. For example, in neuroscience, decoding neural data during learning would be one (the distribution of the input changes as the animal learns the task) or in climate science (the basic models do not accurately forecast weather or temperature as the CO2 emissions increase, for example)
>
> Thanks for your suggestion about CO2 emissions and neuroscience experiments. Although running experiments on more datasets can indeed provide further insights about our method, our paper already covers wide ranges of experiments: synthetic Gaussian, CIFAR-10 and CLEAR continuous supervised learning benchmarks with multiple shifts, RL experiments using 2 challenging environments with multiple settings, and a number of different ablation studies. These comprehensive scenarios provide enough evidence that our method works. That said, we would consider CO2 emissions and neuroscience experiments in future works.

---

> > ### Comment · Reviewer_UiAM · 2022-12-07
> > **Re: Response to Reviewer UiAM**
> >
> > I thank the authors for addressing my comments and for the detailed responses.
> >
> > I now have a better sense of the setting in which this method could be useful ("when the data evolves gradually"), and I think the paper would benefit for a more clear explanation of the setting in which this method could be used (this is a concern shared by other reviewers as well).
> >
> > I agree that there are many synthetic datasets studied in this paper, but still think that a real dataset would greatly increase the quality of the paper, as it would show that this method can be applied in a real setting and how and when to use it.

---

> > > ### Author Response · Authors · 2022-12-07
> > > **Re: Response**
> > >
> > > Thank you again for your feedback.
> > >
> > > We are very glad that our responses adequately addressed your concerns. We'd like to emphasize that only Gaussian experiment uses synthetic dataset and the rest of experiments utilize standard image benchmarks with multiple shifts (i.e. CIFAR-10 and CLEAR) and standard continuous control tasks with multiple settings and baselines (i.e. ROBEL D’Claw and Half-Cheetah WindVel). That being said, your comment about CO2 emissions and neuroscience is well received.
> > >
> > > Finally, we commit to further clarify our setting by releasing the code and updating the text as this requires simple and minor changes to the paper.

---

### Official Review · Reviewer_gsqz · 2022-11-04

**Confidence:** 4
**Correctness:** 4
**Technical Novelty And Significance:** 2
**Empirical Novelty And Significance:** 2
**Recommendation:** 6

**Clarity, Quality, Novelty And Reproducibility:**

The paper is quite clear except for the section 3.2 which I did not fully grasp. The baselines are chosen fairly and well explained and motivated.

**Strength And Weaknesses:**

This paper focuses on the important setting of learning under continual distribution shifts. Indeed data naturally comes as a sequence and waiting until enough data is gathered to marginalize over time can be impractical in many applications. On top of that the authors make a strong case for the weakness of the marginal approach if one is interested in predicting the immediate future. I've found most of the paper to be clearly written; The transition from train/test propensity scores to time dependent ones was easy to follow, even when one isn't familiar with importance weighting and propensity scores. And the formalization of "gradual" shift seems to be coherent and valid in many cases. Therefore using p_T as a proxy for p_T+dt seems reasonable.

However, the experimental setting seems a little limited since 1) it requires access to the whole histore of the data 2) starts training from scratch at each time step. I can hardly imagine a setting where this is feasible in practice or for bigger models / larger datasets. The benefits on the datasets chosen except Gaussian (which might be too toyish) might not justify the overhead of training the discriminative model to estimate importance weights.
The paper focuses on data that evolves through time but also assumes access to multiple samples per time step that are IID. The paper seems to try to be separate from the setting of continual learning but ends up using a sequence of datasets with known boundaries (the timestesp). I'm not 100% sure where the difference lies.

**Summary Of The Paper:**

This paper investigates the setting of training under a gradual distribution shift over time. To mitigate the effect of distribution shift on the performance on future data, they propose applying importance weighting to simulate training under the most recent data distribution. They motivate this by arguing that the latest distribution will be similar to the next future time step since the change in distribution happens gradually. The main contributions of this paper are
- A formalisation of training under distribution shift
- A method to estimate importance weights that depend on time
- Experiments in a supervised and RL setting to showcase the benefits of the method

**Summary Of The Review:**

The setting proposed is promising and an important one to study. The idea presented is simple (in a good way) and well motivated. However, the practical implementation of the algorithm seems to have too much overhead and the chosen benchmark datasets aren't different enough from already well established continual learning settings.

---

> ### Author Response · Authors · 2022-11-18
> **Response to Reviewer gsqz**
>
> Thank you for your feedback. Please also see the main comment above. We hope you will consider increasing your score after seeing our response.
>
> > the experimental setting seems a little limited since 1) it requires access to the whole histore of the data 2) starts training from scratch at each time step. I can hardly imagine a setting where this is feasible in practice or for bigger models / larger datasets. The benefits on the datasets chosen except Gaussian (which might be too toyish) might not justify the overhead of training the discriminative model to estimate importance weights.
>
> We think there is an misunderstanding. We are simply running a large number of experiments---to evaluate the time-varying importance weight estimator at every time t. We do not address the problem of how to update models continually. We are in fact conducting a very large number of experiments (e.g., 100--500 different experiments in Fig 1) and show that the method continues to work no matter the diversity of the past data. To further emphasize, we are not concerned---in this paper---about how to update the model in a continual fashion. We are instead only concerned with how to select past data to train a model that predicts accurately on $p_{T + dt}$.
>
> The issue of how to update the model continually is indeed an important one. But it is something best left for future work and not within the scope of this paper. Now that we have designed a way to select the right past samples to fit a model, we believe that developing methods to continually update models and solve a real-world scenario completely is within reach.
> Let us also note that in many real-world applications, machine learning models are restrained today periodically (e.g., every week or every month). Our paper is expressly designed to understand which data is close to the current distribution and therefore should be used to build a model for the current distribution. Our paper does not address how to update a trained model.
>
> > The paper focuses on data that evolves through time but also assumes access to multiple samples per time step that are IID.
>
> No, we do not need access to large number of samples at each time-step. Even if we have a few data at a time-step, the importance weight estimator will continue to work; it will simply have a larger variance which is partially mitigated by the clipping technique for the score that we use.
> Yes, samples from *each* time-step $t$ need to be IID from $p_t$ to build the importance score. This is a very benign assumption. This does not mean that samples are IID *across* time.
>
> > The paper seems to try to be separate from the setting of continual learning but ends up using a sequence of datasets with known boundaries (the timestesp). I'm not 100% sure where the difference lies.
>
> No, we do not rely on task boundaries. The data in our problem can evolve arbitrarily---but gradually---and the distribution at $p_{t + t’}$ can be the same as that of $p_t$. We need to know the time at which a sample arrives, yes. But this is a very natural assumption and any real-world system that uses data will usually know when the data was recorded.
>
> There is a very clear distinction of what we do and continual learning. We focus on the problem of “how to select which past samples are similar to the current distribution of data”. We use this to predict only the immediate next task, namely $p_\{T+dt\}$. Continual learning focuses on how to update a model to a new task, and it seeks to predict accurately on all past tasks. The two are totally different problems. Further, we have 100s of tasks in our experiments, current continual learning methods are ill-suited to address such a large number of evolving tasks.
>
> > However, the practical implementation of the algorithm seems to have too much overhead and the chosen benchmark datasets aren't different enough from already well established continual learning settings.
>
> Please see our response to your question above about “requiring access to all past data”.
>
> Yes, some of our datasets (e.g., CLEAR benchmark) are also used in the continual learning literature. But the dataset is the only common part and problem settings and setups are very different. Even if we use similar datasets to benchmark and evaluate, we are solving a totally different problem than continual learning. We also develop problem settings with label shifts in contrast to the continual learning literature.

---

> > ### Comment · Reviewer_gsqz · 2022-12-01
> > **Response**
> >
> > Thank you for the clarification! I now understand that the setting is one where we receive sequential chunks of data and the proposed method is to reweigh the past data in order to make it most similar to the current distribution. I better understand the goal and will be increasing my score to 6. I still have some reservations however; you say that the i.i.d assumption on the current chunk of data is benign, but in practice, samples from a given time $t$ are rarely independent even if they might have the same distribution. Imagine for instance doing sentiment analysis on twitter, where the distribution shifts drastically over time: although the tweets at time t might have the same distribution, they are far from being independent since there might be a common cause to many messages (trends or world event). This is somehow mitigated when marginalising over time but can hardly be ignored in the instantaneous case.
> >
> > In any case, I believe the paper would benefit from making the setting more precise: providing concrete examples of where the method can be applied.

---

> > > ### Author Response · Authors · 2022-12-02
> > > **Re: Response**
> > >
> > > Thank you for increasing your score and feedback. We are glad that our responses were helpful and addressed your concerns.
> > >
> > > We agree with the reviewer that there are cases where samples from current time can be rarely independent. However, they are many cases where samples are independent from current time such as online shopping patterns during holiday sessions or how a disease evolves through time. Also, we like to emphasize that samples can be non-IID across time.
> > > Nonetheless, we commit to clarify and address this in the paper.

---

### Official Review · Reviewer_FwNn · 2022-11-06

**Confidence:** 3
**Correctness:** 4
**Technical Novelty And Significance:** 4
**Empirical Novelty And Significance:** 3
**Recommendation:** 6

**Clarity, Quality, Novelty And Reproducibility:**

**Clarity** Clearly written, although it's a bit of a pity Algorithm 1 didn't get a place in the main text.

**Quality** High

**Novelty** Good

**Reproducibility** Presumably sufficient.


**Strength And Weaknesses:**

**Strengths**
* The topic is relevant in practice
* The method is theoretically justified and embedded in a broad context of previous work
* Systematic studies on this topic are still rare, in my opinion.
* The results are convincing

**Weaknesses**
* As far as I understand it, each experiment is repeated only three times, so the statistical significance of the results is somewhat unclear.

**Further suggestions for improvement**
* In some places boldmath should be used so that formula expressions are also written in bold, e.g. Remark 2, 3; Method 1; Figure 2
* Please check if the sentence "The goal in these experiments is to build model that" is really meant that way, or if it should be "a model" or "models".
* Figure 2 partially covers the words "sample cpmplexity" above.
* In the bibliography, the first names of the authors are not written out in some entries, as is the case in the vast majority. E.g. J. Baxter, D.R. Cox.
* Please check the author list for "Alex Krizhevsky, , and Geoffrey Hinton."

**Further comments**
* It would be interesting to apply the method in an example where there is NO shift, i.e., where all data stems from the same distribution and thus the baseline method "Everything" is optimal. Can the proposed method achieve the same performance in this setting?
* The practice of exponentially downweighting older data relative to newer data is not new. Perhaps the authors can clarify this or provide a reference.

**Summary Of The Paper:**

The paper deals with the problem that the behavior of the system to be modeled may change during the time of data acquisition. For the case that the behavior changes only gradually, a method is proposed, theoretically justified and empirically tested, which is intended to achieve good model quality for the time shortly after the end of the data acquisition.

**Summary Of The Review:**

A well-written paper on an important topic that proposes, theoretically justifies, and empirically tests a new method. An increase in significance through a larger number of experiments and an investigation of how the method performs in the case of stationary data is desirable.

---

> ### Author Response · Authors · 2022-11-18
> **Response to Reviewer FwNn**
>
> Thank you for your feedback and suggested experiments. Please also see the main comment above. We hope you will consider increasing your score after seeing our response.
>
>
> > As far as I understand it, each experiment is repeated only three times, so the statistical significance of the results is somewhat unclear.
>
> Per the reviewer’s request, we repeated our supervised experiments of CIFAR-10a/b for our method, Everything, and Fine-tune for 8 different seeds (we increased the number of seeds from 3 to 8) and reported results in Figure 9. As these new experiments show, our results are consistent across 3 and 8 seeds. We also like to emphasize that all our RL experiments are done for 10 seeds which is standard for MuJoCo environments.
>
> > Further suggestions for improvement: Please check if the sentence …, Figure 2 partially covers, In the bibliography…,  check the author list for ,....
>
> Thank you for pointing out those issues. We have updated the paper now and addressed them in the revised version.
>
> > It would be interesting to apply the method in an example where there is NO shift, i.e., where all data stems from the same distribution and thus the baseline method "Everything" is optimal. Can the proposed method achieve the same performance in this setting?
>
> Thanks, this is a great suggestion.  Yes, our method achieves the same performance as Everything. To validate this and per your suggestion, we ran new experiments on the CIFAR-10 benchmark **without** shift. We built continuous classification tasks from this dataset where we used past data to predict future data points. This is the same setting as other supervised learning in the paper except we do not apply any shift to the data. Results of this experiment are shown in Figure 10 of the revised paper. As this experiment shows, our method and Everything perform similarly when there is no shift in the data. This experiment provides further evidence about the applicability of our method and shows that our method works regardless of presence of shifts in the data and it has no negative effect on the performance.
>
> > The practice of exponential downweighting older data relative to newer data is not new. Perhaps the authors can clarify this or provide a reference.
>
> Exponential downweighting of older data is very different from what we do. We estimate a propensity score for the past data and the current data to find past samples that are similar to the ones from the current time-step. Because we use a logistic regression to estimate the propensity score by modeling the evolution of data as an exponential family, the weight-estimator has the form given in Equation (9). This is very different from something like, $w(x, T, t) \propto \exp(-(T-t))$, which is presumably what you mean here.

---

### Official Review · Reviewer_Ytji · 2022-11-07

**Confidence:** 3
**Correctness:** 3
**Technical Novelty And Significance:** 2
**Empirical Novelty And Significance:** 3
**Recommendation:** 3

**Clarity, Quality, Novelty And Reproducibility:**

This draft is clear except for the method part. The novelty of the proposed method is limited while the experiments seem good.

**Strength And Weaknesses:**

Strengths:
- This draft studies a well-motivated problem that is very common in real-world applications.

Weakness:
- The description of the proposed method is not very clear. Section 3 is difficult to follow.
- The proposed method appears to need to first store all the data, then adaptively estimate the weights for all the past data, then optimize. This is not exactly a streaming data learning approach. How does the proposed method differ from offline methods like domain generalization?
- The authors propose two models for estimating $p_t(x)$, the exponential family and deviations from the marginal. Assumptions on the data evolution process are strong. These generational models are not well-discussed. How are these scenarios used, for instance, in what applications? Moreover, these two data generation assumptions do not match the experiments in which the authors use a label shift dataset.
- In the experimental part, the authors use a label shift dataset. It is suggested to compare with the baseline method in Wu et al. (2021).

**Summary Of The Paper:**

This draft studies the problem of continual learning with streaming data in the presence of distribution change. The data distribution is changing over time. The proposed method uses a time-varying importance weight estimator to correct the distribution change. Experiments on online supervised learning and reinforcement learning show the effectiveness of their method.

**Summary Of The Review:**

This draft studies an interesting but important problem in real-world machine learning tasks, especially with the explosion of big data, which are collected over time and the distribution can also change over time. The proposed method is simple, but it lacks novelty. Assumptions on how data evolve are strong and do not accurately reflect reality. Moreover, the proposed method requires the storage of past data, which limits its application to real-world tasks. While the experiments seem good, the datasets used do not match the assumptions made in the method section.

---

> ### Author Response · Authors · 2022-11-18
> **Response to Reviewer Ytji -- Part  1**
>
> We thank you for your review and feedback. We are disappointed that you viewed the paper negatively compared to the other reviews. We hope that, through this rebuttal process, you might view the paper in a new light and increase your score to something more positive.
>
> > The proposed method appears to need to first store all the data, then adaptively estimate the weights for all the past data, then optimize. This is not exactly a streaming data learning approach.
>
> Our method is applicable to the case where data is continuously collected from a constantly evolving distribution such that the single train/test paradigm no longer applies and the model learns and makes predictions based on already seen data. In the case of streaming data, we would have to choose a specific frequency to update the weight-estimator and the trained model. But this is usually a very standard choice, e.g., a lot of models in the real world are updated every week/every month even though data clearly arrives every instant. This is the exact setting we seek to address in our work.
>
> > How does the proposed method differ from offline methods like domain generalization?
>
>  In our paper, we make predictions on future data that are slightly different from recent data. Hence, we will evaluate the test data from “one time-step in the future” and the learner does not have access to any training data from future. This fits into the definition of domain generalization with the caveat that we assume data changes **gradually** not **rapidly** and data are **not** out-of-distribution during test time. Due to this assumption, which is very relevant for real-world scenarios, we can model the patterns in the change in the data distribution across time more precisely (as opposed to domain generalization methods which do not always model this) and obtain better predictions.
>
> > The authors propose two models for estimating pt(x), the exponential family and deviations from the marginal. Assumptions on the data evolution process are strong. These generational models are not well-discussed. How are these scenarios used, for instance, in what applications? Moreover, these two data generation assumptions do not match the experiments in which the authors use a label shift dataset.
>
> We clarify that we neither assume data need to be generated according to specific distribution nor we have any assumptions about relation between exponential family and data distribution. We only assume data changes gradually which happens in real-world events (e.g. Callaway, 2020 ).  Exponential family  (Pitman 1936, Koopman 1936, Wainwright, et al., 2008) is only used to build our time-varying estimator and it has nothing to do with how data is generated. If such an assumption was restrictive, our method would not have worked in the experiments where data distribution clearly evolves differently (like you said). Finally, we added new experiments comparing two models for estimating our method. Figure 5 demonstrates these new results.
>
> [Callaway, 2020]. The coronavirus is mutating — does it matter?
>
> [Pitman, 1936]  Sufficient statistics and intrinsic accuracy.
>
> [Koopman, 1936 ]  On distributions admitting a sufficient statistic
>
> [Wainwright, et al., 2008]  Graphical models, exponential families, and variational inference.
>
> > In the experimental part, the authors use a label shift dataset. It is suggested to compare with the baseline method in Wu et al. (2021).
>
> While we have followed Wu et al. (2021) to generate the labelshift dataset, our data generation is different and more complex than Wu et al. (2021). In particular,  Wu et al. (2021) only consider two classes (Dogs and cats) in their label shift experiments whereas we consider all 10 classes and apply more challenging shifts. Also, Wu et al. (2021) proposed an approach that works only in the presence of label shifts in online learning; however, our approach is not specific to supervised learning or online learning and can be combined with any methods, take our reinforcement learning as an example.
>
> > The proposed method is simple, but it lacks novelty.
>
> We are glad that the reviewer finds our method simple. We value the simplicity of our proposed approach and the simplicity of our method should not detract from its contribution. This is a feature not a bug.

---

> > ### Author Response · Authors · 2022-11-18
> > **Response to Reviewer Ytji -- Part 2**
> >
> > > Assumptions on how data evolve are strong and do not accurately reflect reality. While the experiments seem good, the datasets used do not match the assumptions made in the method section.
> >
> > We have no assumptions on the evolution of data, other than that it evolves slowly. The reviewer is concerned that our propensity score is computed by assuming that data belongs to an exponential family (joint across features x and time t). First, experiments in the paper clearly deal with the case when data does not belong to the exponential family---and they work. So evidently the method does not need such an assumption. Second, the exponential family is dense in the space of distributions; any distribution is close to some member of the exponential family. We only use the propensity score to select past samples that are likely under the current distribution.  The experimental evaluations clearly show that our method works.
> >
> > > Moreover, the proposed method requires the storage of past data, which limits its application to real-world task
> >
> > We think there is a major misunderstanding. We are simply running a large number of experiments---to evaluate the time-varying importance weight estimator at every time t. We do not address the problem of how to update models continually. We are in fact conducting a very large number of experiments (e.g., 100--500 different experiments in Fig 1) and show that the method continues to work no matter the diversity of the past data. To further emphasize, we are not concerned---in this paper---about how to update the model in a continual fashion. We are instead only concerned with how to select past data to train a model that predicts accurately on $p_\{T + dt\}$.
> >
> > The issue of how to update the model continually is indeed an important one. But it is something best left for future work and not within the scope of this paper. Now that we have designed a way to select the right past samples to fit a model, we believe that developing methods to continually update models and solve a real-world scenario completely is within reach.
> > Let us also note that in many real-world applications, machine learning models are restrained today periodically (e.g., every week or every month). Our paper is expressly designed to understand which data is close to the current distribution and therefore should be used to build a model for the current distribution. Our paper does not address how to update a trained model.

---

### Official Review · Reviewer_LTNU · 2022-11-07

**Confidence:** 4
**Clarity, Quality, Novelty And Reproducibility:** good
**Correctness:** 3
**Technical Novelty And Significance:** 3
**Empirical Novelty And Significance:** 2
**Recommendation:** 5

**Strength And Weaknesses:**

Strength: 1. it will be very convenient if we are able to evaluate the contribution of a data point to a new task without knowing exactly which distribution it were sampled from.
 2. the idea of using a DL method for importance weight estimating is interesting.

Weaknesses:
1. the theoretical properties are lacking - The issue of how the variation of the estimator is influenced by the data and how to controll the variation is not discussed in the paper.

2. there are many other methods to do the same thing (importance evaluating), e.g., using influence functions - some clarifying about the difference to those methods is needed.

3.  the supervised learning experiments are mostly based on simulated data without real data shifting -  something like recommendation applications may be considered where people's interests are changing constantly.

4. for the RL part - since the data is depended on what the underlying policy is, I don't think that learning a weighting network without considering the policy that generate the data is very appropriate - those policies are not necessarily changing graduately.

**Summary Of The Paper:**

This paper presents a method to automatically evaluate the contribution of a data point from some historical distribution to a new task without knowing the exact distributions of both old data and the new task. Experiental results show that the proposed wighting method works consitently better than several baseline methods on some simulated data sets.

**Summary Of The Review:**

This paper proposes a neural network to estimate the importance sampling weight. But large variance is one critical issue of any importance sampling weight estimators, so it's better to further consider the effect of variance when using the proposed methods. Another point is that more realistic illustration of the effect of the proposed method is lacking.

---

> ### Author Response · Authors · 2022-11-18
> **Response to Reviewer LTNU**
>
>  Thank you for your valuable feedback. We hope you will consider increasing your score after reading our responses. Please let us know if there are more questions.
>
> > there are many other methods to do the same thing (importance evaluating), e.g., using influence functions - some clarifying about the difference to those methods is needed
>
> There are many ways to estimate the influence of different samples on the estimator, influence functions being one of them. However, in our setting, we cannot use influence functions because even if we were to estimate the influence of a past datum (and this is itself very difficult for neural networks and also because there are 1000s of data points in the past), we would not have any way (other than using the propensity) to select or unselect one of the past data. To our understanding, influence functions are primarily used to study how the model depends upon different samples in the dataset; marshaling such techniques to build models that select the right samples is an open research problem and has never been done before (e.g., there is an exponential number of samples that could be chosen). We are happy to add such a comment to the camera-ready manuscript in the Discussion section.
>
> On the other hand, our weight estimator is a simple tool to select relevant past samples and it can be applied to a wide range of settings (e.g. like RL and supervised learning).
>
>
> > the supervised learning experiments are mostly based on simulated data without real data shifting - something like recommendation applications may be considered where people's interests are changing constantly.
>
>  Thanks for your suggestion about the recommender system. While using more datasets can indeed provide further perspective about our method, our paper already covers wide ranges of experiments: synthetic Gaussian, CIFAR-10 and CLEAR continuous supervised learning benchmarks with multiple shifts, RL experiments using 2 challenging environments with multiple settings, and a number of different ablation studies. These comprehensive scenarios provide enough evidence that our method works. Applying our method to recommender systems which suffer from drift in the data can be a future research direction, but that is beyond the scope of this paper.
>
> > for the RL part - since the data is depended on what the underlying policy is, I don't think that learning a weighting network without considering the policy that generate the data is very appropriate - those policies are not necessarily changing graduately.
>
> That is a good point. While it is correct that the replay buffer (in an off-policy method) contains data from various behavior policies, the beauty of our method is that it models how data evolve over time with respect to the current policy without requiring access to the behavior policies. This makes it very useful in real-world problems where the data are often collected from unknown policies (e.g. offline reinforcement learning). The fact that our method continues to work even when data do not always evolve gradually is a merit of the method.
>
> > But large variance is one critical issue of any importance sampling weight estimators, so it's better to further consider the effect of variance when using the proposed methods.
> the theoretical properties are lacking - The issue of how the variation of the estimator is influenced by the data and how to controll the variation is not discussed in the paper.
>
> Thanks for highlighting this point. The main goal of using an importance weight estimator was to build an unbiased estimator. The weight estimator can have high variance estimates, e.g., we would have seen poor generalization to test data and worse performance than doing no correction at all---but we have not noticed these things in our experiments. A simple variance reduction technique like weight clipping just works well with our method. Since data is changing gradually, the effective sample size of the last time step is large relative to the recent data (small effective sample size means data changing rapidly).
> As our results show, incorporating our time-varying importance weight estimator makes a difference despite having some variance. Finally, it is worth noting that our method can be easily combined with more advanced variance reduction techniques such as doubly robust estimators, etc if required.

---

> > ### Comment · Reviewer_LTNU · 2022-12-10
> > **Thanks for your response.**
> >
> > However, I would like to keep my score unchanged due to the following reasons:
> > 1. The experiments are not very convincing - it does not show how this method works in real life data and for RL applications, it does not show that it did yield state of the art offline RL performance.
> > 2. I think that the proposed method relies too much on the classifer trained to distinguish samples from different distributions.  Algorithm 2 GenerateData shows that the classifier is trained on some augumented data but how much extra data should be generated and how the final performance of the whole method is sensitive to this is not given. I think that it would be useful if some examples were visulized regarding to how samples from time t are useful to the model on time T.
> >
> > In summary, I think that this method is convenient but I'm worry about its performance especially in complicated cases when capturing the distribution similarity between two time steps becomes difficult using the proposed data augumentated classifier.

---

> > > ### Author Response · Authors · 2022-12-10
> > > **Re: Thanks for your response.**
> > >
> > > Thank you for your engagement. It appears that there may have been a misunderstanding about RL experiments and how we train our method. We hope our below comments clarify those for you.
> > >
> > > > The experiments are not very convincing - it does not show how this method works in real life data
> > >
> > > We respectfully disagree with the reviewer and believe that our comprehensive scenarios provide enough evidence that our method works. Our experiments cover both RL and supervised settings where standard image benchmarks, continuous control tasks, and synthetic dataset were used. **It is rare to see a paper to cover both RL and supervised settings in such a thorough way to show the efficacy of a method**. We'd like to highlight that this is a very fresh problem setting (when data evolves gradually and that helps us address distribution shifts in a neat way). We are among the first to formulate this setting and propose an effective solution for it.
> > >
> > > > RL applications, it does not show that it did yield state of the art offline RL performance.
> > >
> > > We are quite puzzled by the above statement. First of all, **we do not have any experiments in this paper for offline RL**. If the reviewer refers to the off-policy RL experiments as offline RL, we are not sure either what the reviewer meant by the state-of-the-art here. Can you please elaborate? As we mentioned in the paper, we combine our time-varying importance estimator with SAC and SAC + Context in off-policy settings and in all experiments our method outperforms these methods ( the only difference between our method and baseline methods is the introduction of time-varying importance estimator term in the Q-update and all other details are exactly the same). We also emphasize that SAC + Context is a strong method for non-stationary settings in RL (Caccia et al., 2022, Ni et al., 2022, and Fakoor et al., 2020) and leads to state-of-art performance in those settings.
> > >
> > > [Caccia et al., 2022] Task-agnostic continual reinforcement learning: In praise of a simple baseline
> > >
> > > [Fakoor et al., 2020] Meta-q-learning.
> > >
> > > [Ni et al., 2022] Recurrent Model-Free RL Can Be a Strong Baseline for Many POMDPs
> > >
> > >
> > > > I think that the proposed method relies too much on the classifer trained to distinguish samples from different distributions. Algorithm 2 GenerateData shows that the classifier is trained on some augumented data but how much extra data should be generated and how the final performance of the whole method is sensitive to this is not given.
> > >
> > > It appears that there may have been a misunderstanding around how the classifier is trained. The way we trained our classifier is a standard way to build a two sample test using a binary classifier. In particular, our Algorithm is similar to how to build a binary classifier for the standard propensity score where the only difference is the introduction of time in our method. Please see section 3 of Agarwal  et al., 2011. Finally, we do not use any extra data to train the classifier and we only use all available training data at a given time-step to train the classifier.
> > >
> > > [Agarwal  et al., 2011 ] Linear-Time Estimators for Propensity Scores
> > >
> > > > I think that it would be useful if some examples were visulized regarding to how samples from time t are useful to the model on time T.
> > >
> > > Per reviewer's request, we include a visualization here [ https://imgur.com/3kBfrtR ] of how our method scores samples across times with respect to the current time for CIFAR10 benchmark. In this figure, scores are shown with respect to their similarity to time T (i.e. T=198) where higher score means more similar. As this figure **clearly** shows, our method can effectively identify samples across times which are similar to the current time step. We'll include this figure in the final version of the paper.
> > >
> > > > In summary, I think that this method is convenient but I'm worry about its performance especially in complicated cases when capturing the distribution similarity between two time steps becomes difficult using the proposed data augumentated classifier.
> > >
> > > In fact, our experiments show the opposite of this statement. If this above statement was correct, our method would not have worked in the diverse set of experiments where data distribution clearly evolves differently in complicated ways in both RL and supervised settings. Besides, one important property of our method is that it is not specific to a particular class of problems and can be utilized in all kinds of settings from supervised learning to reinforcement learning with a combination of existing approaches.

---

> > > > ### Comment · Reviewer_LTNU · 2022-12-10
> > > > **Re：Thanks for your response**
> > > >
> > > > 1. about offline RL / off-policy RL
> > > > sorry that I'm not very clear about this in my previous comment. As the main point of this paper is that the proposed new scheme to caculate the likelihood ratio would work better than the traditional one, I think that a better way to check this is to compare the two types of weighting schemes side by side, instead of adding the proposed weighting factor onto the SAC and showing that this leads to better performance than the standard SAC algorithm (in my impression SAC as an off-policy RL method does not employ the important sampling technique in its standard version).  As an even more interesting testbed,  offline RL, where distribution shifting is one of the major challenges, it would be good if some experimental evidence is provided showing that  the proposed method works well in that case.
> > > >
> > > > 2. about Alg. 2 (i.e., on page15 - Algorithm 2 GenerateData)
> > > > As there is no detailed explanation about how this algorithm works in the text, for me these codes show that to generate a training sample, the algorithm first randomly samples a data point, then either randomly adds a contrasting time stamp onto it to construct a positive sample or randomly replaces its original time stamp with a new one to get a negative sample. Hence theoretically, one can generate as many training samples like this as he/she likes, especially the negative ones.  Please correct me if my understanding is wrong and  my question is, how would this procedure inflence the performance of the likelihood ratio estimator?

---

> > > > > ### Author Response · Authors · 2022-12-10
> > > > > **Re: Re：Thanks for your response**
> > > > >
> > > > > Thank you.
> > > > >
> > > > > > about offline RL / off-policy RL sorry that I'm not very clear about this in my previous comment. As the main point of this paper is that the proposed new scheme to caculate the likelihood ratio would work better than the traditional one, I think that a better way to check this is to compare the two types of weighting schemes side by side, instead of adding the proposed weighting factor onto the SAC and showing that this leads to better performance than the standard SAC algorithm (in my impression SAC as an off-policy RL method does not employ the important sampling technique in its standard version).
> > > > >
> > > > > We indeed compared the standard propensity with our method in Figure 4. As Figure 4 shows, our method fairly detects shifts across different time steps, whereas standard propensity scoring largely ignores shifts across different time steps. While this is done for the supervised learning setting, the same conclusion applies to the RL setting. We will add an ablation study to further show this.
> > > > >
> > > > > One major issue with standard propensity in RL (and equally in supervised learning) is what part of the replay buffer will be considered off-policy data and what part of it will be considered on-policy data when building the binary classifier. This problem becomes even harder as data keeps changing and data which was considered on-policy data in the previous step becomes off-policy data in new time-step (with respect to the current policy). That is likely one of the reasons (if not the main reason) why the standard propensity score is not widely adopted in the off-policy setting. However, our method does not suffer from this issue as it organically depends on time and we don't need to worry about what is off-policy data and what is on-policy data. Importantly, the beauty of our method is that it models how data evolves over time with respect to the current policy **without** requiring access to the behavior policies.
> > > > >
> > > > > > As an even more interesting testbed, offline RL, where distribution shifting is one of the major challenges, it would be good if some experimental evidence is provided showing that the proposed method works well in that case.
> > > > >
> > > > > Yes, offline RL can be an interesting use case to consider. Although distribution shift is an important issue in offline RL, the majority of offline RL method focuses on how to handle the issue of extrapolation error as extrapolation error makes offline RL a challenging problem, see Fujimoto, et. al 2019. While the extrapolation error sometimes can be the result of distribution shift, the main reason for extrapolation in offline RL is due to learning from limited data as it leads to out-of-distribution state-actions during training as data is limited. We hypothesize that propensity score can be effective in offline RL as long as extrapolation error can be effectively addressed somehow. However, addressing extrapolation error in offline RL is not the focus of this paper and outside of scope of this paper.
> > > > >
> > > > > Note that while we didn't consider offline RL in this paper, we show the efficacy of our method in both standard online RL and non-stationary online RL settings and our method outperforms the baseline methods in both cases.
> > > > >
> > > > > [ Fujimoto, et. al 2019 ] Off-Policy Deep Reinforcement Learning without Exploration
> > > > >
> > > > > > about Alg. 2 (i.e., on page15 - Algorithm 2 GenerateData) As there is no detailed explanation about how this algorithm works in the text, for me these codes show that to generate a training sample, the algorithm first randomly samples a data point, then either randomly adds a contrasting time stamp onto it to construct a positive sample or randomly replaces its original time stamp with a new one to get a negative sample. Hence theoretically, one can generate as many training samples like this as he/she likes, especially the negative ones. Please correct me if my understanding is wrong and my question is, how would this procedure inflence the performance of the likelihood ratio estimator?
> > > > >
> > > > > While we agree with the reviewer that Algorithm 2 can use more explanation in the appendix, it is implemented exactly as described in Algorithm 2, it has a very simple and straightforward implementation.
> > > > > It seems the reviewer asks about sensitivity to the value of N in line 4 of Algorithm 2 and how choice of N affects the quality of estimation? This is a fair question. We did a minimal random search (e.g. 2- 3 runs) to select this value and we didn't see sensitivity to the choice of N as long as it is not too small. Note that this value can be even selected automatically by stopping the training after getting to a certain binary classification accuracy.
> > > > >
> > > > >
> > > > > Finally, we hope you still consider increasing your score as we diligently addressed your questions and importantly your concerns don't point to major issues with the paper but rather minor issues.

---

### Author Response · Authors · 2022-11-18
**Response to all reviewers**

We thank the reviewers for their feedback that we've used to greatly improve the paper. We have responded to the concerns of the reviewers as individual comments below.

We are glad that the reviewers found that we study an interesting and well-motivated problem that is relevant in terms of real-world applications (LTNU, Ytji, FwNn, gsqz, UiAM, TuzH). They have said that a systematic study of the problem where the data evolves in time is rare (FwNn). Our method is simple and straightforward (gsqz, TuzH, Ytji) and the paper generally is well-written and presents a clear message (gsqz, FwNn,UiAM, TuzH) that the approach is applicable to different settings (TuzH) and that the main ideas and assumptions presented in the paper are sound (TuzH, LTNU).

To summarize our contributions:

   1. We propose a simple yet effective time-varying importance weight estimator to account for gradual shifts in data distributions;
   2. One important property of our method is that it can be utilized in many different kinds of settings---from supervised learning to reinforcement learning---and can be used in tandem with many existing approaches for these problems;
  3. Our results on a wide range of different settings further highlight the value and effectiveness of our proposed method.

Additional new experiments suggested by the reviewers.

 1. We repeated our experiments on some of continuous supervised learning benchmarks using 8 random seeds for different baseline methods; these (see Fig 9 in the Appendix) show that our conclusions are consistent across the number of random seeds. We also like to emphasize that all our RL experiments in MuJoCo are conducted for 10 seeds.
2. For ROBEL D'Claw, we compared our method with a baseline for the setting in which all methods used context variables (Figure 7 in the appendix).
3. We compared various settings (i.e. Method 1 vs Method 2) of our proposed approach (Figure 5 in the appendix) to show that results are consistent no matter what formulation of the drift estimator we use.
4. We compared our method with training on “Everything” and show that they perform similarly when there is *no* shift in data (Figure 10 in the appendix).

These new results provide additional data points and further show the efficacy and broad applicability of our method. We included these new results in the revised paper.

We also made minor updates to the paper to address some of the reviewers’ comments and uploaded a new version. Please see our responses below that mentioned what have been updated in the text.

---

### Decision · Program_Chairs · 2023-01-20

**Decision:**

Reject

**Justification For Why Not Higher Score:**

As described in the meta-review.

**Justification For Why Not Lower Score:**

none.

**Metareview: Summary, Strengths And Weaknesses:**

The paper poses an approach to continuous learning, a topic that deserves ample attention since it's one where ML systems are notoriously poor in. It addresses this experimentally by varying importance weights.

The authors are applauded for their extensive rebuttal efforts.  Yet, despite its strong experimental section, with experiments both with supervised learning (with data distributions evolving in time) and RL, the paper lacks a thorough theoretical background.  A paper in this form is not sufficiently consequential.